# DRiFT: Differentiable Grid-Based Rigid-Fluid Coupling for Training and Control

## Abstract

Intelligent agents, interacting with physical environments, require an accurate understanding of the consequences of their action for efficient learning. Such agents are often trained inside simulated environments to alleviate over dependence on data, and gradients from such a simulation can help in training the agent. To this end, we present an end-to-end differentiable grid-based fluid simulation including strong two-way coupling with rigid bodies. In the forward pass, the solid-fluid boundary conditions are converted to a monolithic linear pressure solve using a variational method. For the backpropagation, we introduce a novel method of calculating and propagating gradients for the combined fluid-solid state using the adjoint method, which runs faster than the forward solve. This implementation, which is customized for coupling rigid bodies with inviscid fluids, is more suitable over general purpose methods like automatic differentiation, for use cases where performance is key for analyzing overall flow patterns and learning fluid properties. We demonstrate the utility of our simulator in training a neural network to learn optimal control for general target states. Additionally, we show the effectiveness of our differentiable simulator in isolation, by using the generated gradients for simple derivative based optimization tasks. Finally, we showcase the accuracy, robustness and efficiency of our gradient computation method.

## 1 Introduction

There is a rising trend of using ML methods to model physical laws and learn complex interactions in dynamical systems (Sanchez-Gonzalez et al., 2020; Li et al., 2021; Greydanus et al., 2019). These find applications in estimating physical parameters, guiding physical systems in control problems, learning system dynamics for fast inference or to model environment for training RL agents (Ha & Schmidhuber, 2018). For this purpose, increasingly complicated models (Hafner et al., 2024; Kang et al., 2025) have been introduced, which rely on astronomical amounts of data for training, mainly because many physical systems (e.g. fluids) tend to be very high dimensional in nature. Specialised models like PINNs (Raissi et al., 2019), tend to perform better on scarce data, but their physically constrained loss terms are ultimately soft constraints, leading to issues like violation of physical laws or limited numerical accuracy. Constantly increasing training times and resource consumption aside, if these models are to be used for prediction purposes, they would require novel data, especially covering rare phenomena, at a rate far exceeding the rate at which the data is collected in reality.

Due to these challenges, intelligent models are often trained inside environments simulated using physical models, combined with accurate numerical methods, where new data can be generated on-the-fly (Banerjee et al., 2023). Additionally, if the simulators are differentiable, they can provide gradients of objective functions grounded in the environment w.r.t. the physical state or control inputs. These gradients can serve as continuous reward signals for efficient training (Lutter et al., 2020; Heiden et al., 2020; Xian et al., 2023). Apart from improving sample efficiency, these gradients can help explore unintuitive trajectories, which are not obvious with unstructured rewards in traditional RL methods, which usually incorporate few agent behaviours like proximity to objective, contraint violation, etc. Differentiable simulators can also be effective in isolation, for simple derivative based optimization tasks. Consequently, there has been a growing interest in differentiable programming.

There have been significant efforts in developing differentiable simulators for different dynamical systems, including fluids, but relatively few focus on coupling fluids with dynamic obstacles. In

Figure 1: **Differentiable Simulator as an Environment Model.** An agent observes the env. state $q$ and interacts through an action $a$ (such as control force). Depending on how close the final env. state is to the objective $\mathcal{L}$, the simulator generates a reward $\frac{\partial \mathcal{L}}{\partial a}$ which can be used to train the agent. $\Phi$ is the forward pass and $\Psi$ is the adjoint pass. Refer to Figure 2 for details.

fact, because of dynamic boundary conditions, even forward simulation is challenging, making the problem of finding gradients even more challenging. It is important to consider rigid-fluid coupling as it models agent-environment interaction. So, we present an end-to-end differentiable fluid simulator which supports strong two-way rigid-fluid coupling. We employ a grid-based discretization using a MAC grid (Harlow & Welch, 1965). Although this discretization doesn't provide arbitrary accuracy, less degrees of freedom ensure incompressibility and rigid body constraints, and stability in gradient computation. Additionally, our gradient computation is faster than the forward solve, which makes our approach highly scalable. Our main contributions are summarized as follows:

1. We analytically derive the gradients of the entire simulation pipeline, consisting of fluid velocity advection (Section 4.1), monolithic pressure solve for solid-fluid boundary conditions (Section 4.2), rigid body update (section 4.3) and velocity correction (Section 4.4).

2. We demonstrate the application of our computed gradients in an optimization process aimed at estimating initial state of the system, so that it naturally evolves into a desired final state.

3. We utilize our simulator for optimal control, both in isolation for trajectory optimization, and in conjunction with a neural network, for generalization across target states

4. In Section 6.3, we provide validation experiments to demonstrate accuracy and robustness of our simulator. Additionally, we provide comparisons with an auto-differentiation framework PhiFlow to highlight the efficiency of our adjoint based gradient computation method.

## 2 RELATED WORK

Differentiable simulators have been proposed for many dynamical systems, as reviewed comprehensively by Newbury et al. (2024). They offer exciting new learning-enabled applications like structure identification and discovery (Ingraham et al., 2019; Wang et al., 2020), policy and planning (Mora et al., 2021; Xu et al., 2022) as well as design and fabrication (Huang et al., 2021; Nava et al., 2022). Nzoyem Ngueguin et al. (2023) demonstrate how differentiable programming can solve optimal control under PDEs like Laplace equation or Navier Stokes equations, more effectively than PINNs.

Increasing requirements of differentiability of programs in general have spawned sophisticated automatic differentiation(AD) (Baydin et al., 2018) tools, which directly differentiate programs, e.g. by using source code transforms (Hu et al., 2020) or computational graphs (Paszke et al., 2019; Macklin, 2022). PhiFlow (Holl & Thuerey, 2024) and JAX-SPH (Toshev et al., 2024) are frameworks for solving grid-based and particle-based Navier Stokes equation respectively, both of which use AD to compute gradients. Ramos et al. (2022) used PhiFlow, augmented with tailored loss functions, to train a deep neural network controller for a two-way coupled rigid-fluid system. But AD tools are general-purpose and entail overhead, in the form of message passing, maintaining complete gradient info., etc. (Kotary et al., 2023).

While many methods address fluid control without computing gradients of the objective (Pan & Manocha, 2017; Tang et al., 2021), we take inspiration from McNamara et al. (2004), who compute the simulation gradients of level-set based liquids. Takahashi et al. (2021) extend this to support one-way solid to liquid coupling. Li et al. (2024) further extend the pipeline with derivatives of boundary geometry, unlocking optimization based design applications. Two-way coupling additionally considers force transmitted to the solid from the fluid, which is very important for agents interacting with fluid based environments. Li et al. (2023) designed a differentiable simulator for particle-based

(Lagrangian) fluids coupled with rigid bodies, but they report instability in the computation of full analytic gradient. Lagrangian methods adopt a unified particle-based representation for fluids and solids, which simplifies treatment of different entities, but suffer from issues like high computation costs because of more degrees of freedom, failing to preserve fluid incompressibility or rigid body shape constraints. Recent differentiable simulators like FluidLab (Xian et al., 2023), DaXBench (Chen et al., 2023) and Rewarped Xing et al. (2025) use hybrid Lagrangian-Eulerian methods like MPM (Jiang et al., 2016) because of its versatility in handling different continuum materials. However, MPM often cannot enforce strict rigidity because of particle representation, or incompressibility on the liquid, because of explicit pressure computation from constitutive equation, rather than obtaining it from a global projection. So, rigid-fluid interaction is reduced to explicit momentum updates, making it prone to stability issues. This makes MPM unsuitable for strong two-way coupling.

Our main contribution is a differentiable simulator for grid-based (Eulerian) fluids coupled with rigid bodies, which employs far less degrees of freedom, thus promising less resource comsumption and better stability. We highlight a recent work by Lee et al. (2023), who have made a similar contribution as us. However, there are two important distinctions. First, they use immersed boundary method(IBM) (Peskin, 2002) for fluid structure interaction, while our work is inspired by the variational interpretation of pressure by Batty et al. (2007), with a sub-grid accurate discretization using volume fractions which accounts for partially occupied grid cells. This prevents artifacts common to IBM, like stair-steps or fluid seeping through the solid. Secondly, we use the adjoint method (Lions, 1971) for gradient computation which reduces a linear system in a matrix of unknowns to that of a vector of unknowns, thereby accelerating the backward pass.

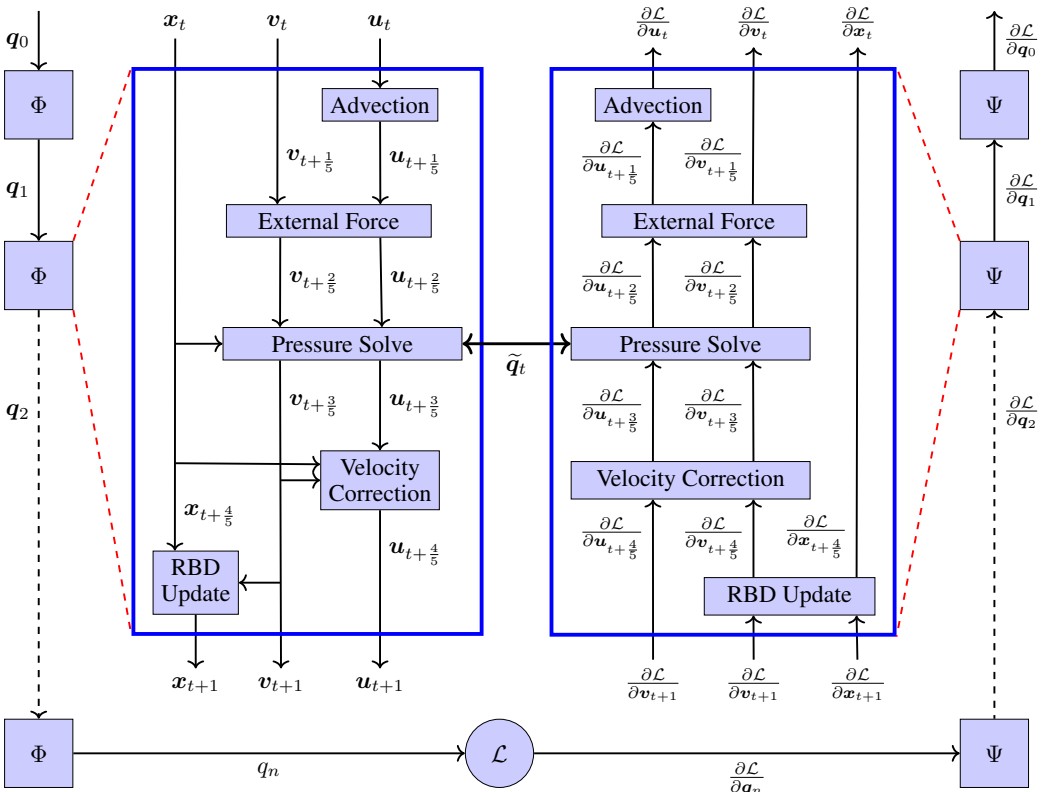

Figure 2: **Algorithm Flow Diagram** shows the forward (left) and backward pass (right) of our differentiable simulator. Each time step acts on the combined solid-fluid state $q$ or the combined adjoint state $\frac{\partial \mathcal{L}}{\partial q}$. The zoomed in diagram shows the sub-stages in one time-step. There is some degree of sharing in each time-step (i.e. $\tilde{q}_t$) between the passes for efficiency. This framework is general and can solve a variety of optimization and learning problems with accurate fluid-solid coupling.

## 3 BACKGROUND

### 3.1 FORWARD SIMULATION

Fluid dynamics are governed by inviscid Navier Stokes equation with the incompressibility constraint:

$$\frac{\partial \boldsymbol{u}}{\partial t} = -\boldsymbol{u} \cdot \nabla \boldsymbol{u} + \frac{1}{\rho}\boldsymbol{f} - \frac{1}{\rho}\nabla p, \quad \nabla \cdot \boldsymbol{u} = 0, \tag{1}$$

where $\boldsymbol{u}$: fluid velocity, $\rho$: fluid density, $p$: fluid pressure and $\boldsymbol{f}$: external force (i.e. gravity, wind forces, etc.) per unit volume. The numerical solution is computed using operator splitting, where each term on the right-hand-side is solved sequentially in writing order. The fluid is discretized as a MAC grid (Harlow & Welch, 1965). Rigid body motion is handled using semi-implicit time integration:

$$\boldsymbol{v}_{t+1} = \boldsymbol{v}_t + \Delta t\, \mathbf{M}_S^{-1}\boldsymbol{F}, \quad \boldsymbol{x}_{t+1} = \boldsymbol{x}_t + \Delta t\, f_x(\boldsymbol{x}_t, \boldsymbol{v}_{t+1}), \tag{2}$$

where $\boldsymbol{v}$: solid velocity, $\boldsymbol{x}$: solid position, $\boldsymbol{F}$: external force (including control forces), and $\mathbf{M}_S$: mass matrix, all of them at centre of mass. $f_x$ is a per element function on vectors, such that $f_x(x, v) = v$, except for quaternions, in which case $f_x(q, \omega) = 0.5([0, \omega] \otimes q)$, $\otimes$ being quaternion multiplication.

Pressure enforces incompressibility and rigid-fluid boundary conditions on the fluid and provides an external force for the rigid bodies, thus acting as the coupling mechanism between the two domains.

Let $\boldsymbol{q}_t = (\boldsymbol{u}_t, \boldsymbol{v}_t, \boldsymbol{x_t})$ represent the state of the combined rigid-fluid system and $\boldsymbol{c}_t$ represent the control force through which an agent interacts with the system being simulated, both at time $t$. Then, if $\Phi$ represents the function corresponding to the numerical simulation, then $\boldsymbol{q}_{t+1} = \Phi(\boldsymbol{q}_t, \boldsymbol{c}_t)$. Refer to Figure 2 for an overview of the simulation pipeline.

### 3.2 ADJOINT SIMULATION

Our objective is to find the optimal initial state $\widetilde{\boldsymbol{q}}_0$ and set of control forces $\widetilde{\boldsymbol{c}}$, collectively denoted as the control variables $\widetilde{\boldsymbol{cv}}$, so that the system results in a desired final state $\widetilde{\boldsymbol{q}}_n$ after $n$ time steps. We pose this as the optimization problem: $\widetilde{\boldsymbol{cv}} = \arg\min_{\boldsymbol{cv}} \mathcal{L}(\boldsymbol{q}_n, \widetilde{\boldsymbol{q}}_n)$, where the loss function $\mathcal{L}$ measures the distance between two states and $\boldsymbol{q}_n = \Phi^n(\boldsymbol{cv})$. Unlike neural networks, we do not need to specialize the loss function or constrain the optimization to avoid violating any boundary conditions, since the simulation based output $\boldsymbol{q}_n$ already enforces them. Because of this, we can use the gradient $\frac{\partial \mathcal{L}}{\partial \boldsymbol{cv}} = \left(\frac{\partial \mathcal{L}}{\partial \boldsymbol{q}_0}, \frac{\partial \mathcal{L}}{\partial \boldsymbol{c}}\right)$ to reach the optimal value $\widetilde{\boldsymbol{cv}}$.

Starting from $\frac{\partial \mathcal{L}}{\partial \boldsymbol{q}_n}$, which can be computed directly from $\mathcal{L}$, we use reverse mode differentiation to obtain $\frac{\partial \mathcal{L}}{\partial \boldsymbol{q}_0}$. For state $\boldsymbol{q}_t$, consider its adjoint state $\boldsymbol{Q}_t = \frac{\partial \mathcal{L}}{\partial \boldsymbol{q}_t} = \left(\frac{\partial \mathcal{L}}{\partial \boldsymbol{u}_t}, \frac{\partial \mathcal{L}}{\partial \boldsymbol{v}_t}, \frac{\partial \mathcal{L}}{\partial \boldsymbol{x}_t}\right)$. From chain rule, $\boldsymbol{Q}_t = \Psi(\boldsymbol{Q}_{t+1})$, where $\Psi = \frac{\partial \Phi}{\partial \boldsymbol{q}}$. Similar to the forward pass, we can extend this to multiple time steps: $\frac{\partial \mathcal{L}}{\partial \boldsymbol{q}_0} = \boldsymbol{Q}_0 = \Psi^n(\boldsymbol{Q}_n) = \Psi^n\left(\frac{\partial \mathcal{L}}{\partial \boldsymbol{q}_n}\right)$. Depending on how the control force $\boldsymbol{c}_t$ affects the current state $\boldsymbol{q}_t$ (which depends on the experiment), the corresponding derivative $\boldsymbol{C}_t = \frac{\partial \mathcal{L}}{\partial \boldsymbol{c}_t}$ can be obtained directly from the adjoint state $\boldsymbol{Q}_t$.

In short, the forward pass $\Phi$ transforms an initial state $\boldsymbol{q}_0$ to $\boldsymbol{q}_n$, while the adjoint pass $\Psi$ transforms the adjoint state $\boldsymbol{Q}_n$ to $\boldsymbol{Q}_0$, closing one loop of optimization. In the next section, we provide details of derivative calculations of the non-trivial stages of the numerical algorithm $\Phi$.

## 4 DIFFERENTIABLE TWO-WAY SOLID-FLUID COUPLING

Constant external forces only add an offset to the velocity, so there is no change in the adjoint state $\boldsymbol{Q}$. A control force $\boldsymbol{c}_t$ at a time step $t$ alters the state as $\boldsymbol{q}' = \boldsymbol{q}_t + \mathbf{B}\boldsymbol{c}_t$ for some matrix $\mathbf{B}$ dependant on the experiment, then $\frac{\partial \mathcal{L}}{\partial \boldsymbol{c}_t} = \mathbf{B}^T \frac{\partial \mathcal{L}}{\partial \boldsymbol{q}'} = \boldsymbol{Q}'$. So, depending on the time step where the control force is applied, its derivative can be obtained from the corresponding adjoint state. In the following subsection, we describe advection, pressure solve, velocity correction and rigid body time integration stages of the numerical algorithm, and their adjoint variants. More details can be found in Figure 2.

## 4.1 ADVECTION

In grid-based methods, it is crucial to account for the movement of fluid while tracking changes in fluid velocity, which remains stationary in space with the background grid, unlike the fluid itself. The advection term $\boldsymbol{u} \cdot \nabla \boldsymbol{u}$ in Equation 1 captures this behavior. To solve the term, we use semi-Lagrangian discretization (Stam, 1999) in space for its stability, and forward Euler discretization in time, for its relatively simple derivative calculation. Bridson et al. (2006) elucidate this concept.

If $\mathcal{I}(\boldsymbol{g}, \boldsymbol{y})$ returns the multilinear interpolation of grid $\boldsymbol{g}$ at location $\boldsymbol{y}$ in space and $\boldsymbol{x}_g$ is a sample location, then $u_a(\boldsymbol{x}_g) = \mathcal{I}(\boldsymbol{u}_t, \boldsymbol{x}_g - \Delta t \mathcal{I}(\boldsymbol{u}_t, \boldsymbol{x}_g))$, where $u_a$ is the advected velocity. Equivalently,

$$\boldsymbol{u}_a = \mathbf{W}\boldsymbol{u}_t, \quad \therefore \frac{\partial \mathcal{L}}{\partial \boldsymbol{u}_t} = \left(\mathbf{W} + \frac{\partial \mathbf{W}}{\partial \boldsymbol{u}}\boldsymbol{u}_t\right)^T \frac{\partial \mathcal{L}}{\partial \boldsymbol{u}_a}, \tag{3}$$

where $\mathbf{W}(\boldsymbol{u}_t)$ is interpolation weight matrix. Row denotes weights corresponding to backtraced location $\boldsymbol{x}_g - \Delta t \mathcal{I}(\boldsymbol{u}_t, \boldsymbol{x}_g)$ inside the grid $\boldsymbol{u}_t$, while column represents weights of a $\boldsymbol{u}_t$ sample.

## 4.2 PRESSURE SOLVE

Pressure enforces incompressibility in fluid interior ($\Omega_F$), solid-fluid impenetratibility constraint on solid-fluid boundary ($\Omega_{FS}$) and ghost-fluid boundary condition (Gibou et al., 2002) on free surface ($\Omega_{FA}$). For a rigid body ($\Omega_S$), it behaves as an external force. Essentially, pressure $p$ satisfies:

$$\begin{aligned}
\boldsymbol{u}_{t+1}(\boldsymbol{y}) &= \boldsymbol{u}_a(\boldsymbol{y}) - \frac{\Delta t}{\rho}\nabla p(\boldsymbol{y}), \quad \boldsymbol{y} \in \Omega_F \\
\nabla \cdot \boldsymbol{u}_{t+1}(\boldsymbol{y}) &= 0, \quad \boldsymbol{y} \in \Omega_F \\
p(\boldsymbol{y}) &= 0, \quad \boldsymbol{y} \in \Omega_{FA} \\
\boldsymbol{v}_{t+1} &= \boldsymbol{v}_t + \Delta t \mathbf{M}_S^{-1}\boldsymbol{F}_p(\boldsymbol{x}_t, p) \\
(\boldsymbol{u}_{t+1}(\boldsymbol{y}) - \boldsymbol{v}_{t+1}(\boldsymbol{y})) \cdot \hat{n} &= 0, \quad \boldsymbol{y} \in \Omega_{FS},
\end{aligned} \tag{4}$$

where $\boldsymbol{q}_t = (\boldsymbol{u}_a, \boldsymbol{v}_t, \boldsymbol{x}_t)$: input state, $\boldsymbol{q}_{t+1} = (\boldsymbol{u}_{t+1}, \boldsymbol{v}_{t+1})$: output state and $\boldsymbol{F}_p$: external force on rigid body. We use the variational interpretation of pressure, inspired from Batty et al. (2007), to convert the set of equations 4 into a monolithic linear system. We rewrite the pressure field as vector $\mathbf{p}$ and the gradient operator as matrix $\mathbf{G}$. Pressure force being linear in pressure, we write $\boldsymbol{F}_p$ as the vector $\mathbf{J}\mathbf{p}$, where matrix $\mathbf{J}$ depends on the location $\boldsymbol{x}_t$ of the rigid body. Treating each velocity sample as a binary decision of presence/absence of liquid in each cell causes artifacts on non-grid aligned boundaries, as investigated previously by Batty et al. (2007); Takahashi & Lin (2019). So, we use a fluid mass matrix $\mathbf{M}_F$ to associate a mass value with each fluid velocity sample, depending on the amount of fluid in the corresponding grid cell. It allows us to properly enforce mass conservation on the fluid. Leaving out the full derivation, which can be found in Bridson et al. (2006), the matrix version of Equations 4 resolves to:

$$(\boldsymbol{u}_{t+1}, \boldsymbol{v}_{t+1}) = \left(\boldsymbol{u}_a - \frac{\Delta t}{\rho}\mathbf{G}\mathbf{p}, \ \boldsymbol{v}_t + \Delta t \mathbf{M}_S^{-1}\mathbf{J}\mathbf{p}\right), \tag{5}$$

where pressure $\mathbf{p}$ is obtained from the linear system:

$$\mathbf{A}\mathbf{p} = \frac{1}{\rho}\mathbf{G}^T\mathbf{M}_F\boldsymbol{u}_a - \mathbf{J}^T\boldsymbol{v}_t, \text{ where } \mathbf{A} = \left[\frac{\Delta t}{\rho^2}\mathbf{G}^T\mathbf{M}_F\mathbf{G} + \Delta t \mathbf{J}^T\mathbf{M}_S^{-1}\mathbf{J}\right]. \tag{6}$$

The corresponding adjoint update is (full derivation in Appendix B):

$$\frac{\partial \mathcal{L}}{\partial \boldsymbol{u}_t} = \frac{\partial \mathcal{L}}{\partial \boldsymbol{u}_{t+1}} + \frac{1}{\rho}\mathbf{M}_F\mathbf{G}\mathbf{s}, \quad \frac{\partial \mathcal{L}}{\partial \boldsymbol{v}_t} = \frac{\partial \mathcal{L}}{\partial \boldsymbol{v}_{t+1}} - \mathbf{J}\mathbf{s}$$

$$\frac{\partial \mathcal{L}}{\partial \boldsymbol{x}_t} = \frac{\partial \mathcal{L}}{\partial \boldsymbol{x}_{t+1}} + \Delta t \left[\mathbf{M}_S^{-1}\left(\frac{\partial \mathbf{J}}{\partial \boldsymbol{x}_t}\right)\mathbf{p}\right]^T \frac{\partial \mathcal{L}}{\partial \boldsymbol{v}_{t+1}} + \Delta t \left[\left(\frac{\partial \mathbf{M}_S^{-1}}{\partial \boldsymbol{x}_t}\right)\mathbf{J}\mathbf{p}\right]^T \frac{\partial \mathcal{L}}{\partial \boldsymbol{v}_{t+1}}$$

$$- \boldsymbol{v}_t^T\left(\frac{\partial \mathbf{J}}{\partial \boldsymbol{x}_t}\right)\mathbf{s} - \Delta t \left[\mathbf{M}_S^{-1}\left(\frac{\partial \mathbf{J}}{\partial \boldsymbol{x}_t}\right)\mathbf{p}\right]^T \mathbf{J}\mathbf{s} - \Delta t \left[\left(\frac{\partial \mathbf{M}_S^{-1}}{\partial \boldsymbol{x}_t}\right)\mathbf{J}\mathbf{p}\right]^T \mathbf{J}\mathbf{s} \tag{7}$$

$$- \Delta t \left[\mathbf{M}_S^{-1}\mathbf{J}\mathbf{p}\right]^T\left(\frac{\partial \mathbf{J}}{\partial \boldsymbol{x}_t}\right)\mathbf{s} - \frac{\Delta t}{\rho^2}\mathbf{p}^T\mathbf{G}^T\left(\frac{\partial \mathbf{M}_F}{\partial \boldsymbol{x}_t}\right)\mathbf{G}\mathbf{s} + \frac{1}{\rho}\boldsymbol{u}_t^T\left(\frac{\partial \mathbf{M}_F}{\partial \boldsymbol{x}_t}\right)\mathbf{G}\mathbf{s}$$

where the adjoint pressure $\mathbf{s}$ can be obtained from the linear system:

$$\mathbf{A}\mathbf{s} = \left[\Delta t \mathbf{J}^T \mathbf{M}_S^{-1}\left(\frac{\partial \mathcal{L}}{\partial \boldsymbol{v_{t+1}}}\right) - \frac{\Delta t}{\rho}\mathbf{G}^T\left(\frac{\partial \mathcal{L}}{\partial \boldsymbol{u_{t+1}}}\right)\right]. \tag{8}$$

### 4.3 RIGID BODY TIME INTEGRATION

This step updates the position of rigid body, $\boldsymbol{x}_{t+1}$, using velocity $\boldsymbol{v}_{t+1}$: $\boldsymbol{x}_{t+1} = \boldsymbol{x}_t + \Delta t\, f_x\left(\boldsymbol{x}_t, \boldsymbol{v}_{t+1}\right)$. For translational components $f_x(x, v) = v$ and $\left(\frac{\partial \mathcal{L}}{\partial x_t}, \frac{\partial \mathcal{L}}{\partial v_{t+1}}\right) = \left(\frac{\partial \mathcal{L}}{\partial x_{t+1}}, \Delta t\, \frac{\partial \mathcal{L}}{\partial x_{t+1}}\right)$.

For quaternions $f_x(q, \omega) = 0.5([0, \omega] \otimes q)$ and,

$$\begin{aligned}\frac{\partial \mathcal{L}}{\partial q_t} &= \left(I_{4 \times 4} + \frac{\Delta t}{2}\frac{\partial\left([0, \omega_{t+1}] \otimes q_t\right)}{\partial q_t}\right)\frac{\partial \mathcal{L}}{\partial q_{t+1}}\\ \frac{\partial \mathcal{L}}{\partial \omega_{t+1}} &= \left(\frac{\Delta t}{2}\frac{\partial\left([0, \omega_{t+1}] \otimes q_t\right)}{\partial \omega_{t+1}}\right)\frac{\partial \mathcal{L}}{\partial q_{t+1}}.\end{aligned} \tag{9}$$

We provide details regarding the partial derivatives of $([0, \omega_{t+1}] \otimes q_t)$ in Appendix C.

### 4.4 FLUID VELOCITY CORRECTION

Because of the choice of discretization, boundary conditions from equations 4 are not applied to grid cells with no fluid, but their samples may be used during advection in later time steps. These *invalid* velocities must be corrected to make sure that the advected velocity obeys the boundary conditions.

The method of correction depends on where the invalid velocity sample lies. If it lies in the air domain ($\Omega_A$), it can be extrapolated from adjacent valid velocity samples, using Gauss-Seidel type iterations. A larger time step requires larger number of iterations. In our experiments, 3 such iterations work sufficiently well. If on the other hand, the velocity sample lies in the interior of a rigid body ($\Omega_S$), it is set to the appropriate component of the rigid body velocity at the sample point. In both cases, the corrected velocity field $\boldsymbol{u}_c$ is linear in the combined projected velocity $(\boldsymbol{u}_{t+1}, \boldsymbol{v}_{t+1})$:

$$\boldsymbol{u}_c = \mathbf{C}\begin{bmatrix}\boldsymbol{u}_{t+1}\\\boldsymbol{v}_{t+1}\end{bmatrix} \implies \begin{bmatrix}\frac{\partial \mathcal{L}}{\partial \boldsymbol{u}_{t+1}} & \frac{\partial \mathcal{L}}{\partial \boldsymbol{v}_{t+1}}\end{bmatrix}^T = \mathbf{C}^T\left(\frac{\partial \mathcal{L}}{\partial \boldsymbol{u_c}}\right), \tag{10}$$

where $C$ is the correction matrix.

## 5 IMPLEMENTATION

Our differentiable simulator is written in C++ and parallelized using OpenMP. Forward pass is adapted from *FluidRigidCoupling2D* library (Batty, 2016). We use Eigen (Guennebaud et al., 2010) for all linear algebra operations, libigl (Jacobson & Panozzo, 2017) and tetgen (Si, 2015) for all mesh related operations, Takahashi & Batty (2022)'s code for volume fractions and *SplashSurf* library (Löschner et al., 2023) to recontruct liquid surface, before being rendered in Blender. Figure 3 and Appendix A contain the pseudocode and the code is provided in the supplementary. The experiments, written in Python, access the simulator using *pybind* (Jakob et al., 2017). All experiments are run either on an Intel Xeon W-2123 CPU at 3.6 GHz with 8 threads and 32 GB memory or an AMD Ryzen 3700X CPU with 16 threads at 4.1 GHz and 32 GB memory. Check out our website for video results.

## 6 EXPERIMENTS AND EVALUATIONS

### 6.1 INITIAL STATE ESTIMATION

In this experiment, we show the utility of our differentiable simulator in an inverse problem of estimating initial state which results into a desired final state after a fixed number of time steps. We accomplish this through derivative based optimization using gradients from the simulator.

**Dam Break.** We perform this experiment in both 2D and 3D. In 3D, we set up a box domain of size $3.5 \times 2 \times 5$ containing $70 \times 40 \times 100$ grid cells of size 0.05. A ball of relative density 0.3 is

---

**Algorithm 1** Differentiable Simulation

**Global:**

    1. Grid Dimensions, $(Nx, Ny, Nz)$

    2. Grid Cell Width, $\Delta x$

    3. Boundary SDF, $\phi_B$

    4. Rigid Body(ies) Geometry + Mass ($\mathbf{M}_S$)

    5. Constant external forces i.e. gravity, control forces

**Input:** Initial state $\boldsymbol{q}_0 = (\boldsymbol{u}_0, \boldsymbol{v}_0, \boldsymbol{x}_0)$, Initial particles $\boldsymbol{pt}_0$, Number of frames $f$, Time step size $\Delta t$, Loss function $\mathcal{L}$

1: $\boldsymbol{q}, \boldsymbol{pt} \leftarrow \boldsymbol{q}_0, \boldsymbol{pt}_0$
2: **for** $t = 1$ **to** $f$ **do**
3:     Forward Simulation: $\boldsymbol{q}, \boldsymbol{pt}, \widetilde{\boldsymbol{q}} \leftarrow \Phi(\boldsymbol{q}, \Delta t)$
4:     Store results: $\widetilde{\boldsymbol{q}}$
5: **end for**
6: $\boldsymbol{Q} \leftarrow \text{getLossDerivative}(\mathcal{L}, \boldsymbol{q})$
7: **for** $t = f$ **to** $1$ **do**
8:     Adjoint Simulation: $\boldsymbol{Q} \leftarrow \Psi(\boldsymbol{Q}, \boldsymbol{q}_t, \widetilde{\boldsymbol{q}}_t, \Delta t)$
9: **end for**

**Output:** Final adjoint state, $\boldsymbol{Q} = \frac{\partial \mathcal{L}}{\partial \boldsymbol{q}_0}$

---

Figure 3: Pseudocode for our simulator. Forward/Adjoint simulations are detailed in Appendix A

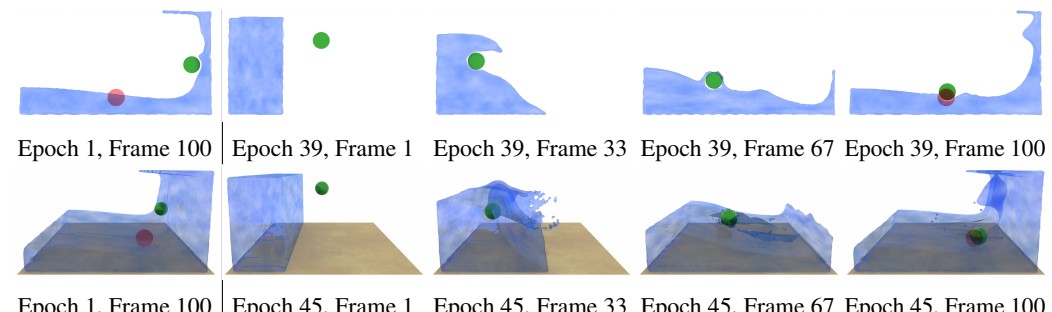

| Epoch 1, Frame 100 | Epoch 39, Frame 1 | Epoch 39, Frame 33 | Epoch 39, Frame 67 | Epoch 39, Frame 100 |

| Epoch 1, Frame 100 | Epoch 45, Frame 1 | Epoch 45, Frame 33 | Epoch 45, Frame 67 | Epoch 45, Frame 100 |

Figure 4: **Dam Break.** We optimize initial velocity of ball(green) so that it reaches the target(red). Epoch 1 starts with a guess, by Epoch 39(2D) and Epoch 45(3D), the ball is quite close to target.

thrown from $(1.7, 1.4, 1.5)$ against the wall of water at the start of the simulation. The objective is to learn the initial velocity with which to throw so that it reaches $(2.4, 0.4, 3.5)$. In 2D, domain size is $3.5 \times 2$ with $70 \times 40$ grid cells of side length $0.05$. The ball starts from $(1.7, 1.4)$ and the target is at $(1.75, 0.5)$. The simulations are performed for $100$ frames ($\Delta t = 0.01$). We use gradient descent with fixed LR of $0.5$ for this experiment. Figure 4 shows that our simulator learns the optimum initial velocity in both cases. Figure 7a shows the loss trends.

**Text Optimization.** We demonstrate our solver's ability to jointly optimize multiple rigid bodies by spelling the word "HI". The simulation domain is $15 \times 15 \times 15$ with $0.1$ m spacing, and runs for $100$ frames at $0.01$ second time steps for $20$ epochs. The Adam Kingma & Ba (2017) optimizer is used with a starting learning rate of $0.5$. The fluid surface starts at $0.6$ meters from the bottom of the domain and all four blocks are dropped from a height of $0.8$ meters. Three long blocks measure $1.5 \times 1.5 \times 4.0$ grid cells; the fourth is $1.5^3$. All of them start with zero initial velocity and the loss is the L2 distance against the desired position not considering rotation. In the first epoch, the blocks hit the water surface and bounce indiscriminately; however by the end of training, the blocks submerge themselves and bounce out to form the word. Results are shown in Fig 5 along with the smoothly decreasing loss curve in Fig 7b.

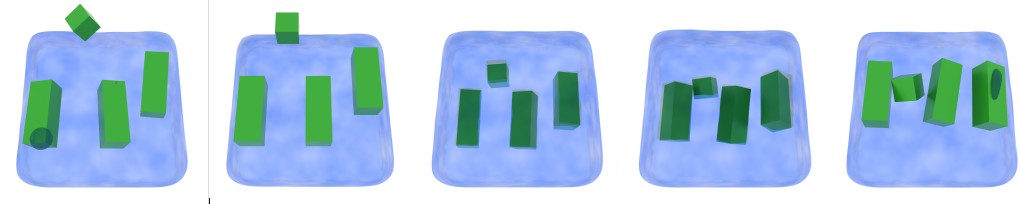

Epoch 1, Frame 100 | Epoch 19, Frame 1 | Epoch 19, Frame 33 | Epoch 19, Frame 66 | Epoch 19, Frame 100

Figure 5: **Text Optimization.** We optimize the final position of four blocks to spell the word "HI" after submerging and being pushed out of water. We see that our simulation is able to determine initial linear velocities capable of reaching the correct final position after complex interactions.

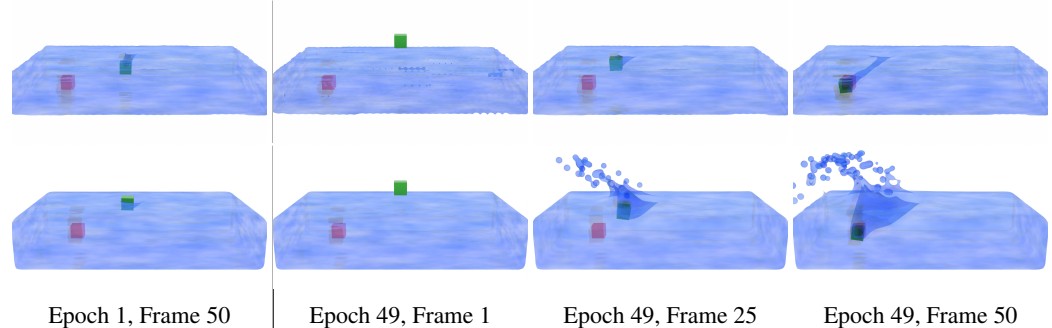

Epoch 1, Frame 50 | Epoch 49, Frame 1 | Epoch 49, Frame 25 | Epoch 49, Frame 50

Figure 6: **Block Drop.** We optimize the position of a single block over a large water surface. L2 loss between the final and target position is used. Our method is able to compute the correct gradients for the block(green) to find the correct initial velocity to reach the desired position(red). The top row shows our result while the bottom row shows results from DiffFR (Li et al., 2023)

**Comparison with DiffFR.** We compare our method with SPH-based differentiable simulator DiffFR Li et al. (2023), using a rigid-body drop experiment. The objective is to optimize the initial velocity of a block so that it reaches a target position and orientation. The domain is box shaped with dimensions $3.5 \times 1.5 \times 2$ and $39 \times 15 \times 24$ grid cells for our method and 237699 particles for DiffFR. The cube, with side length 2 times the grid cell size of $0.1$ and density twice that of the liquid, is initially dropped with velocity $(1, -1, 1)$ onto the liquid covering half of the domain. The time step of the simulation is $5 \times 10^{-3}$ and it is performed for 50 frames, and 50 epochs. Each forward pass was around 6.6 seconds, while the backward pass was about 3.7 seconds. See Fig 6 for results. Both methods converge to a reasonable solution in about 20 epochs, but our method greatly reduces the computational cost. Our simulation runs about **20 times faster** reducing a 2 hour, 51 minute run to 8.6 minutes, due to the differences between the Eulerian and Lagrangian discretizations. Figure 7c shows the loss curves for the two methods.

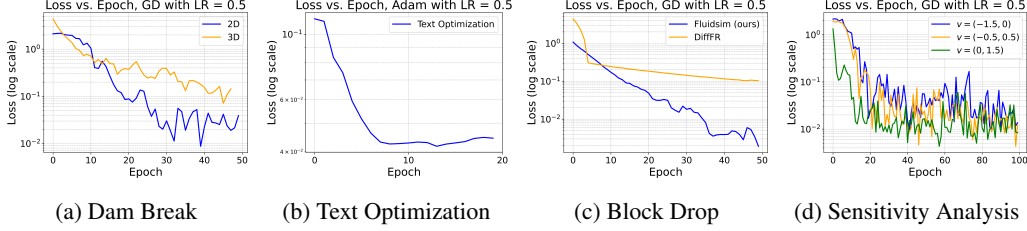

(a) Dam Break | (b) Text Optimization | (c) Block Drop | (d) Sensitivity Analysis

Figure 7: **Loss Curves.** We demonstrate that the gradients calculated by our solver are able to quickly decrease the loss in our experiments. We also compare our simulation against the particle based DiffFR and explore the sensitivity to different initial conditions.

## 6.2 OPTIMAL CONTROL

Since our simulator is differentiable end-to-end, we can also obtain gradients w.r.t. control forces applied at different time steps of the simulation. We use these gradients to train a neural network controller to guide a rigid body towards a given target state, while interacting with the surrounding liquid. The rigid body starts at $(1.0, 1.0)$ in a box domain of size $2 \times 2$ with $40 \times 40$ grid cells filled with water, which has an initial anticlockwise swirling motion with velocity FV. The neural net takes as input, the initial combined solid-fluid state and target state, and generates a control force sequence, which is applied to the rigid body over the next few time steps i.e. horizon H. We study the performance of controller for different values of FV and H. During training, convergence is reached in all cases (Figure 8a). During testing, the controller is applied to 15 target configurations. Figure 8b shows the trajectories taken by the rigid body. Higher swirl velocity in Figure 8c causes the rigid body to curve more, while longer horizon in Figure 8d necessitates indirect trajectories. More importantly, the control forces help the rigid body to move close to the target regardless of the environment state, since the gradients from simulator encode exactly that information.

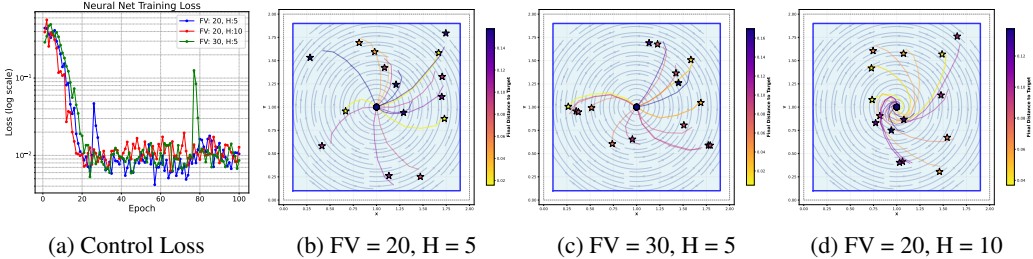

| (a) Control Loss | (b) FV = 20, H = 5 | (c) FV = 30, H = 5 | (d) FV = 20, H = 10 |

Figure 8: **Optimal Control. (a)** shows training loss v/s epoch, where FV is tangential fluid velocity and H is horizon length. **(b), (c)** and **(d)** show rigid body trajectory in test cases. Fluid is shown in light blue, with dark blue domain boundaries. Fluid motion is shown by light grey lines. Dark blue circle at the centre represents initial RBD position. Target locations are denoted by stars, with color matching the corresponding trajectory. Colorbar indicates how close a trajectory reaches the target.

## 6.3 ANALYSIS AND COMPARISONS

**Gradient Accuracy Analysis.** For every stage of the numerical algorithm $\phi_i$, we start from some initial state $q_0$, apply $\phi_i$, $\mathcal{L}$ and $\psi_i$ in order, to compute $\frac{\partial \mathcal{L}}{\partial q_0}$. Then, we perturb $q_0$ by $\delta = 10^{-5}$, one component at a time, in positive and negative directions to obtain $q_1$ and $q_{-1}$. These are used to compute the centered finite difference gradient $(2\delta)^{-1}[\mathcal{L}(\phi_i(q_1)) - \mathcal{L}(\phi_i(q_{-1}))]$, which we use to validate our computed gradient $\frac{\partial \mathcal{L}}{\partial q_0}$ and report our results in Table 1. We perform this experiment in simple 2D and 3D scenarios with a box domain of side length 1m with 20 grid cells in each dimension, with the domain completely submerged in liquid, and one dynamic rigid box of size 3 grid cells, at a height of 0.7m. The target state $q_t$ is set so that the gradient values are significantly larger than the perturbation. For most cases, the computed gradient is very close to the finite difference gradient (at most $1.3\%$ for a single frame, and at most $7.7\%$ for 10 frames), but it increases with number of frames, which we strongly believe is because of accumulation of numerical error from advection.

| Function | 2D | | | 3D | | |
|---|---|---|---|---|---|---|
| | $u$ | $v$ | $x$ | $u$ | $v$ | $x$ |
| Advection ($\phi_1$) | $8.3 \cdot 10^{-1}$ | - | - | $5.5 \cdot 10^{-1}$ | - | - |
| Pressure Solve ($\phi_2$) | $6.5 \cdot 10^{-7}$ | $1.3 \cdot 10^{-4}$ | $7.3 \cdot 10^{-2}$ | $2.9 \cdot 10^{-5}$ | $1.7 \cdot 10^{-8}$ | $3.5 \cdot 10^{-1}$ |
| RBD Update ($\phi_3$) | - | $4.2 \cdot 10^{-10}$ | $1.4 \cdot 10^{-8}$ | - | $4.3 \cdot 10^{-9}$ | $2.3 \cdot 10^{-9}$ |
| Velocity Correction ($\phi_4$) | $5.3 \cdot 10^{-8}$ | $1.1 \cdot 10^{-10}$ | $1.4 \cdot 10^{-9}$ | $1.4 \cdot 10^{-5}$ | $7.8 \cdot 10^{-10}$ | $2.1 \cdot 10^{-9}$ |
| Forward Pass ($\Phi$) | $1.05$ | $1.4 \cdot 10^{-8}$ | $9.7 \cdot 10^{-2}$ | $1.2 \cdot 10^{-2}$ | $1.8 \cdot 10^{-6}$ | $1.93$ |
| 10 frames ($\Phi^{10}$) | $4.82$ | $1.1 \cdot 10^{-1}$ | $6.85$ | $2.74$ | $7.67$ | $7.12$ |

Table 1: This table shows the average % relative error between FD gradient and gradient computed by our simulator. Rows indicate the function and columns indicate the component of the gradient being investigated. We find that for all functions, our computed gradient is very close to the FD gradient.

**Runtime Analysis.** In Table 2, we list the runtimes of the forward simulation and adjoint-based gradient computation algorithm for each of the demos shown in Section 6.1. One prominent point is that the adjoint pass is much faster than the forward pass at lower resolutions, primarily because of marker particle advection, which is crucial for tracking and reconstructing liquid surface. This step takes up a significant proportion of the forward pass runtime ($O(PMT)$ in asymptotic notation). This step is not necessary in the adjoint pass, resulting in the runtime difference. At higher resolutions however, the extra volume fraction computations which are required in the adjoint pass to compute the term $\frac{\partial \mathbf{J}}{\partial \boldsymbol{x}}$ ($O(NM)$ in asymptotic notation) surpass the marker particle advection, making adjoint pass slower (check Appendix G). We further refer the reader to Appendix D for detailed complexity analysis of our algorithm. Additionally, we compare our runtimes with that of PhiFlow (Holl & Thuerey, 2024), which is written in Python and leverages PyTorch (Paszke et al., 2019) for auto-differentiation(AD) (i.e. unrolling). We find that unrolling in PhiFlow is slower than our adjoint-based method at lower resolutions, while PhiFlow is faster at higher resolutions. Refer to Appendices

| Experiment | Forward Pass ($\Phi$) (s) | | Adjoint Pass ($\Psi$) (s) | | |
|---|---|---|---|---|---|
| | Ours | PhiFlow | Ours | PhiFlow | Speedup |
| Dam Break 3D | 27.2 | 83.9 | 18.7 | 101.3 | 5.41 |
| Dam Break 2D | 0.15 | 0.80 | 0.11 | 0.98 | 8.9 |
| Text Optimization | 0.33 | 7.31 | 0.32 | 7.54 | 23.6 |
| Block Drop 2D | 0.06 | 0.35 | 0.04 | 0.39 | 9.75 |
| Block Drop 3D | 6.58 | 31.4 | 3.68 | 36.9 | 10.0 |

Table 2: This table compares the runtimes of a forward and backward pass (in s) of our simulator with PhiFlow in five scenarios. In all cases, our method outperforms PhiFlow.

**Gradient Sensitivity Analysis.** Our simulator is robust to the choice of initial state, and we test this using 2D dam break experiment set up earlier. The ball starts with three different initial velocities in separate runs. The simulations are allowed to run for 100 frames ($\Delta t = 0.01$) and 100 epochs. Figure 7d shows that despite the optimization trajectories being different, the simulator eventually reaches the optima, regardless of the initial velocity guess. Additionally we test the stability of our simulation in Appendix F and the stability of the optimization trajectory in Appendix E

## 7 CONCLUSIONS, LIMITATIONS AND FUTURE WORK

Physical simulation is important in modeling and engineering our world. The solutions to the equations that govern the simulations are incredibly high dimensional and represent complex interactions. By incorporating gradient information through adjoint simulation, first-order optimization of simulation states and parameters becomes feasible, which greatly improves convergence, as compared to black box or hand-tuned optimization. The derivation of this backward pass is non-trivial, and just like the forward pass, is prone to numerical errors depending on the method of simulation used, for the same analytical derivatives. In this work, we have proposed, designed and implemented an efficient fluid simulator which can provide stable gradients.

But our method has some limitations and it can be further improved. Our simulator is grid-based, where accuracy is proportional to the resolution of the grid. Differentiable hybrid methods, which can leverage both grid and particle representations adaptively can find a good tradeoff between accuracy and efficiency. Secondly, our simulator is predominantly CPU parallelized, which limits the speedup of our method at higher resolution grids. Extending current GPU parallelization beyond marker particle operations to grid-based operations can make our method even more suitable to handle higher resolution scenarios. Next, extending the simulator to support derivatives with respect to the fluid signed distance functions would help in a whole new range of design problems. Finally, our method computes derivatives from the numerical procedure of the forward simulation. An alternative way might be to directly differentiate the analytical formulation of solid-fluid coupling to derive adjoint dynamics, and use a different numerical procedure for its treatment. It would be interesting to see how these two approaches of derivative calculation differ.

ETHICS STATEMENT

We acknowledge that we have read the ICLR Code of Ethics and are committed to adhere to it.

REPRODUCIBILITY STATEMENT

In this work, we have made every possible effort to make our claims and results reproducible. First of all, we provide a detailed derivation of our gradient computation method in Appendices B and C. Then, we provide an overview of our complete algorithm through an explanatory diagram (Figure 2) and pseudocode (Appendix A). Additionally, for all our experiments in Section 6.3, we provide settings and parameters required to replicate the results. Finally, we provide our code in the supplementary for direct reproducibility.

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

## A  SIMULATOR PSEUDO CODE

---

**Algorithm 2** Forward Simulation ($\Phi$), One Time Step

---

**Input:** State $q_t$, Marker Particles $pt_t$
1: $pt_{t+1} \leftarrow \text{advectParticles}(pt_t, q_t, \Delta t)$
2: $q_{t+\frac{1}{5}} \leftarrow \text{advectVelocity}(q_t, \Delta t)$
3: $q_{t+\frac{2}{5}} \leftarrow \text{externalForce}(q_{t+\frac{1}{5}}, \Delta t)$
4: $\phi_L \leftarrow \text{computeLiquidSDF}(pt_{t+1})$
5: $q_{t+\frac{3}{5}} \leftarrow \text{solvePressure}(q_{t+\frac{2}{5}}, \phi_L, \Delta t)$
6: $q_{t+\frac{4}{5}} \leftarrow \text{velocityCorrection}(q_{t+\frac{3}{5}}, \Delta t)$
7: $q_{t+1} \leftarrow \text{updateRigidBody}(q_{t+\frac{4}{5}}, \Delta t)$
**Output:** State $q_{t+1}$, Particles $pt_{t+1}$

---

---

**Algorithm 3** Adjoint Simulation ($\Psi$), One Time Step

---

**Input:** Adjoint state $Q_{t+1}$, Storage state $\widetilde{q}_t$
1: $Q_{t+\frac{4}{5}} \leftarrow \text{adjointUpdateRigidBody}(Q_{t+1}, \widetilde{q}_t, \Delta t)$
2: $Q_{t+\frac{3}{5}} \leftarrow \text{adjointVelocityCorrection}(Q_{t+\frac{4}{5}}, \widetilde{q}_t, \Delta t)$
3: $Q_{t+\frac{2}{5}} \leftarrow \text{adjointSolvePressure}(Q_{t+\frac{3}{5}}, \widetilde{q}_t, \Delta t)$
4: $Q_t \leftarrow \text{adjointAdvectVelocity}(Q_{t+\frac{2}{5}}, \widetilde{q}_t, \Delta t)$
**Output:** Adjoint state $Q_t$

---

The storage state $\widetilde{q}$ is used for sharing data between the forward and adjoint passes to avoid recomputation. In our implementation, in addition to storing all the intermediate states $q$ generated in all sub-stages of the forward pass, we also store SDFs of liquid ($\phi_F$) and solid ($\phi_S$), their volume fractions ($\mathbf{VF}_F$, $\mathbf{VF}_S$) and the computed pressure $\mathbf{p}$.

## B  DERIVATION: GRADIENTS OF PRESSURE SOLVE

The pressure solve or projection step transforms the combined input state $(u_t, v_t, x_t)$ into a combined output velocity $(u_{t+1}, v_{t+1})$ that obeys the physical constraints highlighted in eqn. (2). Starting

here, we wish to compute the downstream gradients $\left( \frac{\partial \mathcal{L}}{\partial \boldsymbol{u}_t}, \frac{\partial \mathcal{L}}{\partial \boldsymbol{v}_t}, \frac{\partial \mathcal{L}}{\partial \boldsymbol{x}_t} \right)$, using the upstream gradients $\left( \frac{\partial \mathcal{L}}{\partial \boldsymbol{u}_{t+1}}, \frac{\partial \mathcal{L}}{\partial \boldsymbol{v}_{t+1}}, \frac{\partial \mathcal{L}}{\partial \boldsymbol{x}_{t+1}} \right)$. Let's say we require $\frac{\partial \mathcal{L}}{\partial \boldsymbol{q}}$, where $\boldsymbol{q} = \boldsymbol{u}_t, \boldsymbol{v}_t, \boldsymbol{x}_t$. Then, on differentiating the linear system of eqn. (2),

$$\mathbf{A} \left( \frac{\partial \mathbf{p}}{\partial \boldsymbol{q}} \right) + \frac{\partial \mathbf{A}}{\partial \boldsymbol{q}} \mathbf{p} = \frac{\partial \mathbf{b}}{\partial \boldsymbol{q}}$$

$$\therefore \mathbf{A} \left( \frac{\partial \mathbf{p}}{\partial \boldsymbol{q}} \right) = \frac{1}{\rho} \mathbf{G}^T \mathbf{M}_F \left( \frac{\partial \boldsymbol{u}_t}{\partial \boldsymbol{q}} \right) - \left( \frac{\partial \mathbf{J}}{\partial \boldsymbol{q}} \right)^T \boldsymbol{v_t} - \mathbf{J}^T \left( \frac{\partial \boldsymbol{v}_t}{\partial \boldsymbol{q}} \right) - \Delta t \left( \frac{\partial (\mathbf{J}^T \mathbf{M}_S^{-1} \mathbf{J})}{\partial \boldsymbol{x}_t} \right) \mathbf{p}$$

$$\rightarrow \mathbf{A} \left( \frac{\partial \mathbf{p}}{\partial \boldsymbol{u}_t} \right) = \frac{1}{\rho} \mathbf{G}^T \mathbf{M}_F, \quad \mathbf{A} \left( \frac{\partial \mathbf{p}}{\partial \boldsymbol{v}_t} \right) = -\mathbf{J}^T$$

$$\rightarrow \mathbf{A} \left( \frac{\partial \mathbf{p}}{\partial \boldsymbol{x}_t} \right) = \frac{1}{\rho} \mathbf{G}^T \left( \frac{\partial \mathbf{M}_F}{\partial \boldsymbol{x}_t} \right) \boldsymbol{u}_t - \left( \frac{\partial \mathbf{J}}{\partial \boldsymbol{x}_t} \right)^T \boldsymbol{v_t} - \Delta t \, \mathbf{J}^T \mathbf{M}_S^{-1} \left( \frac{\partial \mathbf{J}}{\partial \boldsymbol{x}_t} \right) \mathbf{p}$$

$$- \Delta t \left( \frac{\partial \mathbf{J}}{\partial \boldsymbol{x}_t} \right) \mathbf{M}_S^{-1} \mathbf{J} \mathbf{p} - \Delta t \, \mathbf{J}^T \left( \frac{\partial \mathbf{M}_S^{-1}}{\partial \boldsymbol{x}_t} \right) \mathbf{J} \mathbf{p} - \frac{\Delta t}{\rho^2} \mathbf{G}^T \left( \frac{\partial \mathbf{M}_F}{\partial \boldsymbol{x}_t} \right) \mathbf{G} \mathbf{p}$$

Now, differentiating the velocity updates of eqn. (2),

$$\frac{\partial \boldsymbol{u}_{t+1}}{\partial \boldsymbol{q}} = \frac{\partial \boldsymbol{u}_t}{\partial \boldsymbol{q}} - \frac{\Delta t}{\rho} \mathbf{G} \left( \frac{\partial \mathbf{p}}{\partial \boldsymbol{q}} \right), \quad \frac{\partial \boldsymbol{v}_{t+1}}{\partial \boldsymbol{q}} = \frac{\partial \boldsymbol{v}_t}{\partial \boldsymbol{q}} + \Delta t \mathbf{M}_S^{-1} \left( \frac{\partial \mathbf{J}}{\partial \boldsymbol{q}} \right) \mathbf{p} + \Delta t \mathbf{M}_S^{-1} \mathbf{J} \left( \frac{\partial \mathbf{p}}{\partial \boldsymbol{q}} \right)$$

Now, we find $\frac{\partial \mathcal{L}}{\partial \boldsymbol{q}}$ in terms of $\left( \frac{\partial \mathcal{L}}{\partial \boldsymbol{u}_{t+1}}, \frac{\partial \mathcal{L}}{\partial \boldsymbol{v}_{t+1}}, \frac{\partial \mathcal{L}}{\partial \boldsymbol{x}_{t+1}} \right)$, using the principle of total derivative,

$$\frac{\partial \mathcal{L}}{\partial \boldsymbol{q}} = \left( \frac{\partial \boldsymbol{u}_{t+1}}{\partial \boldsymbol{q}} \right)^T \frac{\partial \mathcal{L}}{\partial \boldsymbol{u}_{t+1}} + \left( \frac{\partial \boldsymbol{v}_{t+1}}{\partial \boldsymbol{q}} \right)^T \frac{\partial \mathcal{L}}{\partial \boldsymbol{v}_{t+1}} + \left( \frac{\partial \boldsymbol{x}_{t+1}}{\partial \boldsymbol{q}} \right)^T \frac{\partial \mathcal{L}}{\partial \boldsymbol{x}_{t+1}}$$

$$\rightarrow \frac{\partial \mathcal{L}}{\partial \boldsymbol{u}_t} = \left( \frac{\partial \boldsymbol{u}_{t+1}}{\partial \boldsymbol{u}_t} \right)^T \frac{\partial \mathcal{L}}{\partial \boldsymbol{u}_{t+1}} + \left( \frac{\partial \boldsymbol{v}_{t+1}}{\partial \boldsymbol{u}_t} \right)^T \frac{\partial \mathcal{L}}{\partial \boldsymbol{v}_{t+1}} + \left( \cancel{\frac{\partial \boldsymbol{x}_{t+1}}{\partial \boldsymbol{u}_t}}^{\nearrow 0} \right)^T \frac{\partial \mathcal{L}}{\partial \boldsymbol{x}_{t+1}}$$

$$= \left( \frac{\partial \boldsymbol{u}_t}{\partial \boldsymbol{u}_t} - \frac{\Delta t}{\rho} \mathbf{G} \left( \frac{\partial \mathbf{p}}{\partial \boldsymbol{u}_t} \right) \right)^T \frac{\partial \mathcal{L}}{\partial \boldsymbol{u}_{t+1}}$$

$$+ \left( \cancel{\frac{\partial \boldsymbol{v}_t}{\partial \boldsymbol{u}_t}}^{\nearrow 0} + \Delta t \mathbf{M}_S^{-1} \cancel{\left( \frac{\partial \mathbf{J}}{\partial \boldsymbol{u}_t} \right)}^{\nearrow 0} \mathbf{p} + \Delta t \mathbf{M}_S^{-1} \mathbf{J} \left( \frac{\partial \mathbf{p}}{\partial \boldsymbol{u}_t} \right) \right)^T \frac{\partial \mathcal{L}}{\partial \boldsymbol{v}_{t+1}}$$

$$= \frac{\partial \mathcal{L}}{\partial \boldsymbol{u}_{t+1}} + \left( \frac{\partial \mathbf{p}}{\partial \boldsymbol{u}_t} \right)^T \left[ \Delta t \mathbf{J}^T \mathbf{M}_S^{-1} \frac{\partial \mathcal{L}}{\partial \boldsymbol{v}_{t+1}} - \frac{\Delta t}{\rho} \mathbf{G}^T \frac{\partial \mathcal{L}}{\partial \boldsymbol{u}_{t+1}} \right]$$

$$\rightarrow \frac{\partial \mathcal{L}}{\partial \boldsymbol{v}_t} = \left( \frac{\partial \boldsymbol{u}_{t+1}}{\partial \boldsymbol{v}_t} \right)^T \frac{\partial \mathcal{L}}{\partial \boldsymbol{u}_{t+1}} + \left( \frac{\partial \boldsymbol{v}_{t+1}}{\partial \boldsymbol{v}_t} \right)^T \frac{\partial \mathcal{L}}{\partial \boldsymbol{v}_{t+1}} + \left( \cancel{\frac{\partial \boldsymbol{x}_{t+1}}{\partial \boldsymbol{v}_t}}^{\nearrow 0} \right)^T \frac{\partial \mathcal{L}}{\partial \boldsymbol{x}_{t+1}}$$

$$= \left( \cancel{\frac{\partial \boldsymbol{u}_t}{\partial \boldsymbol{v}_t}}^{\nearrow 0} - \frac{\Delta t}{\rho} \mathbf{G} \left( \frac{\partial \mathbf{p}}{\partial \boldsymbol{v}_t} \right) \right)^T \frac{\partial \mathcal{L}}{\partial \boldsymbol{u}_{t+1}}$$

$$+ \left( \frac{\partial \boldsymbol{v}_t}{\partial \boldsymbol{v}_t} + \Delta t \mathbf{M}_S^{-1} \cancel{\left( \frac{\partial \mathbf{J}}{\partial \boldsymbol{v}_t} \right)}^{\nearrow 0} \mathbf{p} + \Delta t \mathbf{M}_S^{-1} \mathbf{J} \left( \frac{\partial \mathbf{p}}{\partial \boldsymbol{v}_t} \right) \right)^T \frac{\partial \mathcal{L}}{\partial \boldsymbol{v}_{t+1}}$$

$$= \frac{\partial \mathcal{L}}{\partial \boldsymbol{v}_{t+1}} + \left( \frac{\partial \mathbf{p}}{\partial \boldsymbol{v}_t} \right)^T \left[ \Delta t \mathbf{J}^T \mathbf{M}_S^{-1} \frac{\partial \mathcal{L}}{\partial \boldsymbol{v}_{t+1}} - \frac{\Delta t}{\rho} \mathbf{G}^T \frac{\partial \mathcal{L}}{\partial \boldsymbol{u}_{t+1}} \right]$$

$$\rightarrow \frac{\partial \mathcal{L}}{\partial \boldsymbol{x}_t} = \left(\frac{\partial \boldsymbol{u}_{t+1}}{\partial \boldsymbol{x}_t}\right)^T \frac{\partial \mathcal{L}}{\partial \boldsymbol{u}_{t+1}} + \left(\frac{\partial \boldsymbol{v}_{t+1}}{\partial \boldsymbol{x}_t}\right)^T \frac{\partial \mathcal{L}}{\partial \boldsymbol{v}_{t+1}} + \left(\underbrace{\frac{\partial \boldsymbol{x}_{t+1}}{\partial \boldsymbol{x}_t}}_{I}\right)^T \frac{\partial \mathcal{L}}{\partial \boldsymbol{x}_{t+1}}$$

$$= \left(\underbrace{\frac{\partial \boldsymbol{u}_t}{\partial \boldsymbol{x}_t}}_{0} - \frac{\Delta t}{\rho}\mathbf{G}\left(\frac{\partial \mathbf{p}}{\partial \boldsymbol{x}_t}\right)\right)^T \frac{\partial \mathcal{L}}{\partial \boldsymbol{u}_{t+1}} + \frac{\partial \mathcal{L}}{\partial \boldsymbol{x}_{t+1}}$$

$$+ \left(\underbrace{\frac{\partial \boldsymbol{v}_t}{\partial \boldsymbol{x}_t}}_{0} + \Delta t \mathbf{M}_S^{-1}\left(\frac{\partial \mathbf{J}}{\partial \boldsymbol{x}_t}\right)\mathbf{p} + \Delta t \mathbf{M}_S^{-1}\mathbf{J}\left(\frac{\partial \mathbf{p}}{\partial \boldsymbol{x}_t}\right) + \Delta t \left(\frac{\partial \mathbf{M}_S^{-1}}{\partial \boldsymbol{x}_t}\right)\mathbf{J}\mathbf{p}\right)^T \frac{\partial \mathcal{L}}{\partial \boldsymbol{v}_{t+1}}$$

$$= \frac{\partial \mathcal{L}}{\partial \boldsymbol{x}_{t+1}} + \Delta t \left[\mathbf{M}_S^{-1}\left(\frac{\partial \mathbf{J}}{\partial \boldsymbol{x}_t}\right)\mathbf{p}\right]^T \frac{\partial \mathcal{L}}{\partial \boldsymbol{v}_{t+1}} + \Delta t \left[\left(\frac{\partial \mathbf{M}_S^{-1}}{\partial \boldsymbol{x}_t}\right)\mathbf{J}\mathbf{p}\right]^T \frac{\partial \mathcal{L}}{\partial \boldsymbol{v}_{t+1}}$$

$$+ \left(\frac{\partial \mathbf{p}}{\partial \boldsymbol{x}_t}\right)^T \left[\Delta t \mathbf{J}^T \mathbf{M}_S^{-1}\frac{\partial \mathcal{L}}{\partial \boldsymbol{v}_{t+1}} - \frac{\Delta t}{\rho}\mathbf{G}^T\frac{\partial \mathcal{L}}{\partial \boldsymbol{u}_{t+1}}\right]$$

For any vector $\mathbf{d}$, the quantity $\left(\frac{\partial \mathbf{p}}{\partial \boldsymbol{q}}\right)^T \mathbf{d}$ given $\mathbf{A}\left(\frac{\partial \mathbf{p}}{\partial \boldsymbol{q}}\right) = \mathbf{B}$ is equivalent to $\mathbf{B}^T \mathbf{s}$ given $\mathbf{A}^T \mathbf{s} = \mathbf{d}$. This is the adjoint method, first introduced for differentiable fluid simulation by McNamara et al. (2004). Combined with the fact that $\mathbf{A}$ is symmetric:

$$\frac{\partial \mathcal{L}}{\partial \boldsymbol{u}_t} = \frac{\partial \mathcal{L}}{\partial \boldsymbol{u}_{t+1}} + \frac{1}{\rho}\mathbf{M}_F \mathbf{G}\mathbf{s},$$

$$\frac{\partial \mathcal{L}}{\partial \boldsymbol{v}_t} = \frac{\partial \mathcal{L}}{\partial \boldsymbol{v}_{t+1}} - \mathbf{J}\mathbf{s}$$

$$\frac{\partial \mathcal{L}}{\partial \boldsymbol{x}_t} = \frac{\partial \mathcal{L}}{\partial \boldsymbol{x}_{t+1}} + \Delta t \left[\mathbf{M}_S^{-1}\left(\frac{\partial \mathbf{J}}{\partial \boldsymbol{x}_t}\right)\mathbf{p}\right]^T \frac{\partial \mathcal{L}}{\partial \boldsymbol{v}_{t+1}} + \Delta t \left[\left(\frac{\partial \mathbf{M}_S^{-1}}{\partial \boldsymbol{x}_t}\right)\mathbf{J}\mathbf{p}\right]^T \frac{\partial \mathcal{L}}{\partial \boldsymbol{v}_{t+1}} - \boldsymbol{v}_t^T\left(\frac{\partial \mathbf{J}}{\partial \boldsymbol{x}_t}\right)\mathbf{s}$$

$$- \Delta t \left[\mathbf{M}_S^{-1}\left(\frac{\partial \mathbf{J}}{\partial \boldsymbol{x}_t}\right)\mathbf{p}\right]^T \mathbf{J}\mathbf{s} - \Delta t \left[\left(\frac{\partial \mathbf{M}_S^{-1}}{\partial \boldsymbol{x}_t}\right)\mathbf{J}\mathbf{p}\right]^T \mathbf{J}\mathbf{s} - \Delta t \left[\mathbf{M}_S^{-1}\mathbf{J}\mathbf{p}\right]^T \left(\frac{\partial \mathbf{J}}{\partial \boldsymbol{x}_t}\right)\mathbf{s}$$

$$- \frac{\Delta t}{\rho^2}\mathbf{p}^T\mathbf{G}^T\left(\frac{\partial \mathbf{M}_F}{\partial \boldsymbol{x}_t}\right)\mathbf{G}\mathbf{s} + \frac{1}{\rho}\boldsymbol{u}_t^T\left(\frac{\partial \mathbf{M}_F}{\partial \boldsymbol{x}_t}\right)\mathbf{G}\mathbf{s}$$

$$\text{where } \mathbf{A}\mathbf{s} = \Delta t \mathbf{J}^T \mathbf{M}_S^{-1}\frac{\partial \mathcal{L}}{\partial \boldsymbol{v}_{t+1}} - \frac{\Delta t}{\rho}\mathbf{G}^T\frac{\partial \mathcal{L}}{\partial \boldsymbol{u}_{t+1}}$$

## C   DERIVATION: GRADIENTS OF QUATERNIONS

We represent 3D orientation using quaternions $q = [w, v]$, where $v = (x, y, z)$. Given angular velocity $\omega = (\omega_x, \omega_y, \omega_z)$ about three principal axes, the quaternion update equation is:

$$q_{t+1} = \frac{\tilde{q}_{t+1}}{||\tilde{q}_{t+1}||}, \text{ normalization}$$

$$\therefore \left(\frac{\partial q_{t+1}}{\partial \tilde{q}_{t+1}}\right)_{ij} = \begin{cases} \frac{1-(q_{t+1})_i^2}{||\tilde{q}_{t+1}||} & i = j \\ \frac{-(q_{t+1})_i(q_{t+1})_j}{||\tilde{q}_{t+1}||} & i \neq j \end{cases}$$

$$\tilde{q}_{t+1} = q_t + \frac{\Delta t}{2}\left([0, \omega_{t+1}] \otimes q_t\right), \text{ time integration}$$

$$= q_t + \frac{\Delta t}{2}[\omega_{t+1} \cdot (q_t)_v, (q_t)_w\,\omega_{t+1} + \omega_{t+1} \times (q_t)_v]$$

$$\therefore \frac{\partial \tilde{q}_{t+1}}{\partial q_t} = I_{4\times4} + \frac{\Delta t}{2} \begin{bmatrix} 0 & -(\omega_{t+1})_x & -(\omega_{t+1})_y & -(\omega_{t+1})_z \\ (\omega_{t+1})_x & 0 & -(\omega_{t+1})_z & (\omega_{t+1})_y \\ (\omega_{t+1})_y & (\omega_{t+1})_z & 0 & -(\omega_{t+1})_x \\ (\omega_{t+1})_z & -(\omega_{t+1})_y & (\omega_{t+1})_x & 0 \end{bmatrix}$$

$$\therefore \frac{\partial \tilde{q}_{t+1}}{\partial \omega_{t+1}} = \frac{\Delta t}{2} \begin{bmatrix} -(q_t)_x & -(q_t)_y & -(q_t)_z \\ (q_t)_w & (q_t)_z & -(q_t)_y \\ -(q_t)_z & (q_t)_w & (q_t)_x \\ (q_t)_y & -(q_t)_x & (q_t)_w \end{bmatrix}$$

## D   COMPUTATIONAL COMPLEXITY ANALYSIS

Suppose the number of grid cells is $N$, number of marker particles is $P$ and there are $M$ rigid bodies with maximum $T$ triangles each. And let the number of frames in an epoch be $F$.

### D.1   MEMORY

#### D.1.1   ANALYSIS

- Fluid State, $O(N+P)$: Velocity, $O(N)$ + Pressure, $O(N)$ + SDF (signed distance function), $O(N)$ + Marker Particles, $O(P)$

- Rigid Body State, $O(M + N)$: Velocity, $O(M)$ + Position, $O(M)$ + SDF, $O(N)$

- Combined State, $O(M + N + P)$

- Sparse Gradient Matrix **G**, $O(N)$

- Solid-Fluid Coupling Matrix **J**, $O(NM)$

- One iteration(frame) of forward pass, $O(MN + P)$

- One iteration(frame) of backward pass, $O(MN)$: No marker particles in backward pass

- One optimization epoch (forward + backward pass), $O(F \times (M + N))$: Because marker particles and temporary matrices are discarded after every frame

#### D.1.2   IMPROVEMENTS

We can save some memory per frame by using sparser **J** to bring down its space to $O(N)$, but this is not significant in a longer simulation since it is discarded anyway at the end of the frame. A more effective startegy would be to not store any intermediate frames, but use **checkpointing** for backward pass. This brings down the memory requirements to $O(M + N)$ but increases the computation time of the backward pass by an equivalent factor.

### D.2   TIME (UNPARALLELIZED)

#### D.2.1   ANALYSIS

- Advection, $O(N)$

- Particle Advection + Fluid SDF Computation, $O(PMT)$

- Rigid Body Update, $O(M)$

- Rigid Body SDF Computation, $O(MT)$

- Pressure Solve, $O(NM + (N + M)^{1.5})$: Matrix constructions, $O(NM)$ + Linear solve using conjugate gradients, $O((N + M)^{1.5})$

- Velocity Correction, $O(N + M)$: sparse matrix-vector multiplication of size $M + N$

- One iteration(frame) of forward pass, $O(NM + (N + M)^{1.5} + PMT)$

- One iteration(frame) of backward pass, $O(NM + (N + M)^{1.5})$

- One optimization epoch (forward + backward pass), $O(F(NM + (N + M)^{1.5} + PMT))$

### D.2.2 IMPROVEMENTS

For a fixed background grid, the above reported times are the best possible times (on a single thread/core) as far as we are aware. In practice, the particle advection (i.e. $O(PMT)$) takes the most time. So, one way is to optimize marker particle advection or employ a grid-based advection scheme like level-set advection. Another way this can be made much faster is if we use **adaptive grids**, which we are currently working on. Adpative grids have finer resolution in areas requiring higher accuracy (i.e. near solid-fluid boundary) and coarser resolution in other macroscopic areas (i.e. fluid regions). Such grids have the potential to be efficient both in space and time. But differentiability is a challenge in adaptive grids because of dynamic size of state variables.

## E  OPTIMIZATION STABILITY

In this section, we analyze the stability of our optimization process by performing the experiments in the paper for longer, specifically 10 times the number of epochs in the main paper, for the demos in Section 6.1. The results are shown in Figure 9. For most of the cases, the optimization trajectory is stable. For Dam Break 2D, the optimization trajectory is oscillatory but the loss stays between $10^{-5}$ and $10^{-1}$, with minimum value about $5.3 \times 10^{-6}$. For text optimization experiment, the loss decreases on average for the first 100 epochs, then increases slightly and decreases again to attain a minimum value of $5 \times 10^{-3}$. We believe the oscillations in the optimization trajectory are more due to the complicated loss landscape in these demos, and the fact that we use gradient descent with a fixed learning rate as optimizer, and less to do with the accuracy of our gradients, which we validated earlier in Section 6.3. We leave the exploration of this loss landscape and the application of better optimizers to future work.

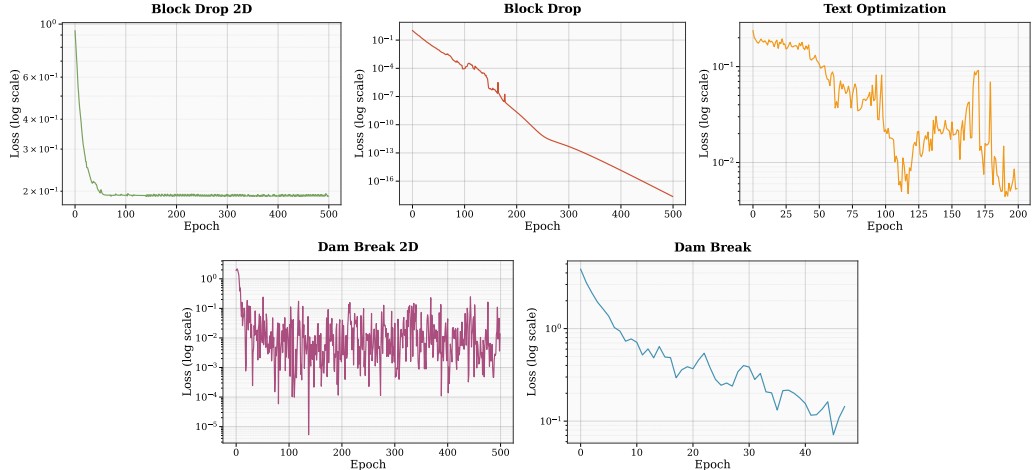

Figure 9: **Optimization Stability.** This experiment extends the demos of Section 6.1 to $10\times$ more epochs than in the main paper to analyze the optimization trajectory. For all the cases, the loss stays low on average. In some cases like Dam Break 2D and Text Optimization, we see oscillations, which we believe might be due to our fixed learning rate optimizer or the complex loss landscape, rather than the accuracy or stability of our solver.

## F  SIMULATION STABILITY

In this section, we test the stability of the our simulator to input perturbations. We take each of the demos of Section 6.1, perturb the initial rigid body position and velocity by varying amounts $\delta$ ranging from $10^{-7}$ to $10^{-3}$, and study the relative change in final simulation state $\boldsymbol{q} = (\boldsymbol{u}, \boldsymbol{x}, \boldsymbol{v})$. Figure 10 contains the plots of the instability factor $\frac{\Delta \boldsymbol{q}}{\delta}$, for each of the components.

The simplest demo, block drop, which contains a single rigid body interacting with static liquid, has the lowest instability factor which is fairly constant for any value of input perturbation $\delta$. Dam

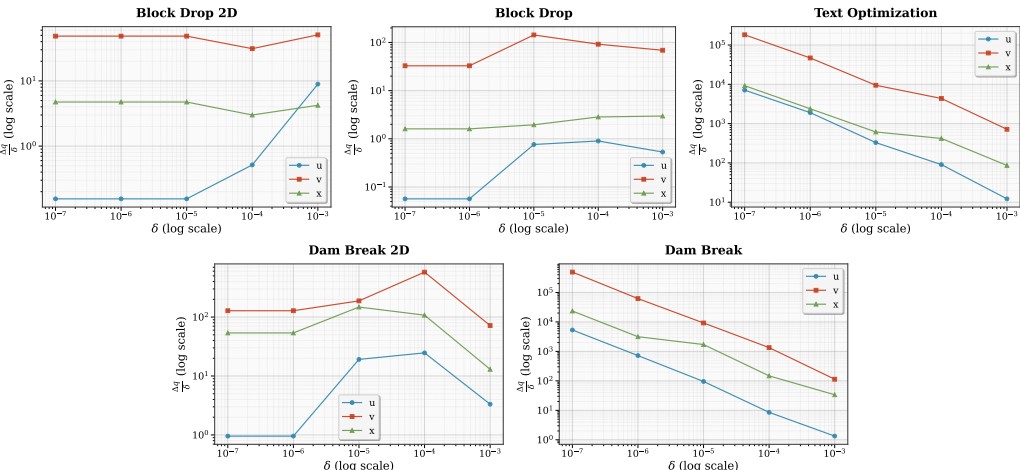

Figure 10: **Simulation Stability Plots.** Each plot shows the instability factor: the ratio of change in output $\Delta q$ to the change in input $\delta$. The ratio is higher the more complicated the demo is, with least in block drop and highest in text optimization. $\Delta q$ is obtained by taking an average over individual perturbations $\Delta q_i$ resulting from pertubing a single input component of the initial state. An individual perturbation is obtained from the L2 norm between the perturbed and the original output.

Break scenario, which is more complicated and involves dynamic liquid has a higher instability. Finally, the text optimization scenario has large instability factor at extremely small perturbations. This may be because the perturbed rigid body might be clipping the external boundary or another rigid body, which is not handled by our simulator, atleast not differentiably, at the moment. **Note that these instability plots do not mean that our simulator is unstable, because the actual output perturbations are quite small in value, as shown in Table 3**. It's just that the relative values are large at small input perturbations, which might be because of the minimum inherent inaccuracy in grid-based discretization.

| $\delta$ | $10^{-3}$ | | | $10^{-4}$ | | | $10^{-5}$ | | | $10^{-6}$ | | | $10^{-7}$ | | |
|---|---|---|---|---|---|---|---|---|---|---|---|---|---|---|---|
| Exp. | $\Delta u$ | $\Delta x$ | $\Delta v$ | $\Delta u$ | $\Delta x$ | $\Delta v$ | $\Delta u$ | $\Delta x$ | $\Delta v$ | $\Delta u$ | $\Delta x$ | $\Delta v$ | $\Delta u$ | $\Delta x$ | $\Delta v$ |
| Dam Break 3D | $1.3e-3$ | $3.4e-2$ | $1.1e-1$ | $8.6e-4$ | $1.5e-2$ | $1.3e-1$ | $9.5e-4$ | $1.7e-2$ | $9.2e-2$ | $7.2e-4$ | $3.1e-3$ | $6.2e-2$ | $5.3e-4$ | $2.3e-3$ | $4.2e-2$ |
| Dam Break 2D | $3.3e-3$ | $1.3e-2$ | $7.2e-2$ | $2.5e-3$ | $1.1e-2$ | $5.8e-2$ | $1.9e-4$ | $1.5e-3$ | $1.9e-3$ | $9.6e-7$ | $5.4e-5$ | $1.3e-4$ | $9.6e-8$ | $5.4e-6$ | $1.3e-5$ |
| Text Optimization | $1.2e-2$ | $8.6e-2$ | $7.2e-1$ | $9.1e-3$ | $4.2e-2$ | $4.4e-1$ | $3.3e-3$ | $6.1e-3$ | $9.5e-2$ | $1.9e-3$ | $2.4e-3$ | $4.7e-2$ | $7.1e-4$ | $9.2e-4$ | $1.8e-2$ |
| Block Drop 2D | $8.9e-3$ | $4.2e-3$ | $5.1e-2$ | $5.1e-5$ | $3.0e-4$ | $3.1e-3$ | $1.6e-6$ | $4.7e-5$ | $4.9e-4$ | $1.6e-7$ | $4.7e-3$ | $4.9e-5$ | $1.6e-8$ | $4.7e-7$ | $4.9e-6$ |
| Block Drop 3D | $5.3e-4$ | $3.0e-3$ | $6.9e-2$ | $9.0e-5$ | $2.8e-4$ | $9.2e-3$ | $7.6e-6$ | $2.0e-5$ | $1.4e-3$ | $5.7e-8$ | $1.6e-6$ | $3.3e-5$ | $5.7e-9$ | $1.6e-7$ | $3.3e-6$ |

Table 3: **Output Perturbations:** This table shows the output perturbations $\Delta q$ corresponding to the stability plots in Figure 10. Note that the values decrease in general with decreasing input perturbations, but the rate of decrease, as summarized in Figure 10, is lower for more complex scenarios, like text optimization and dam break

## G    SCALABILITY ANALYSIS

Our differentiable simulator can support higher resolution simulations. We demonstrate this by performing the 2D Dam Break (Figure 4, top row) experiment on different resolution grids. In each of the runs, we adjust the grid size $\Delta x$ so that the size of the domain is fixed to $3.5 \times 2$. We also change the time step $\Delta t$ by the same factor as $\Delta x$ to obey the CFL condition, and accordingly increase the number of frames $F$ to keep the simulation length at $1s$. The runtimes are listed in Table 4. Our method is faster at lower resolutions, with the speedup decreasing at higher resolutions. PhiFlow surpasses our method in performance at $280 \times 160$ grid resolution, which we believe might be because of their better GPU parallelization.

At lower resolutions, the adjoint pass is faster than the forward pass because the bottleneck is the marker particle advection, which takes up a significant portion of the runtime. But at higher resolutions, the matrix construction step for $\frac{\partial \mathbf{J}}{\partial \boldsymbol{x}}$ in the adjoint step takes much more time than marker particle advection. In terms defined in Appendix $D$, $O(MN)$ surpasses $O(PMT)$ at higher values of $N$. So, adjoint pass is slower at higher resolutions.

| Resolution | $\Delta x$ | $\Delta t$ | F | Forward Pass ($\Phi$) (s) | | Adjoint Pass ($\Psi$) (s) | | |
|---|---|---|---|---|---|---|---|---|
| | | | | Ours | PhiFlow | Ours | PhiFlow | Speedup |
| $35 \times 20$ | $10^{-1}$ | $2 \times 10^{-2}$ | 50 | $5.05 \times 10^{-2}$ | $4.8 \times 10^{-1}$ | $2.13 \times 10^{-2}$ | $5.16 \times 10^{-1}$ | 24.22 |
| $70 \times 40$ | $5 \times 10^{-2}$ | $10^{-2}$ | 100 | $1.74 \times 10^{-1}$ | $7.86 \times 10^{-1}$ | $1.21 \times 10^{-1}$ | $9.57 \times 10^{-1}$ | 7.91 |
| $140 \times 80$ | $2.5 \times 10^{-2}$ | $5 \times 10^{-3}$ | 200 | 1.36 | 2.05 | 1.49 | 2.71 | 1.82 |
| $280 \times 160$ | $1.25 \times 10^{-2}$ | $2.5 \times 10^{-3}$ | 400 | 4.30 | 6.59 | 18.46 | 9.72 | 0.53 |

Table 4: This table compares the runtimes of a forward and backward pass (in s) of our simulator with PhiFlow at five resolution settings of 2D Dam Break (Figure 4). We find that our method is significantly faster at lower resolutions but the speedup decreases at higher resolutions.

