# OpenReview forum: "DRiFT: Differentiable Grid-Based Rigid-Fluid Coupling for Training and Control"
_ICLR.cc/2026/Conference — Submitted to ICLR 2026_

### Official Review · Reviewer_Nvep · 2025-10-26

**Soundness:** 2
**Presentation:** 2
**Contribution:** 1
**Rating:** 2
**Confidence:** 3

**Summary:**

This paper introduces DRiFT, a grid-based differentiable simulator designed to handle strong two-way coupling between rigid bodies and inviscid fluids. The primary contribution is a complete, end-to-end differentiable pipeline. In the forward pass, solid-fluid boundary conditions are unified into a monolithic linear pressure solve using a variational method. For the backward pass, the authors derive a novel adjoint method to efficiently back-propagate gradients for the combined fluid-solid state. They claim this adjoint pass is faster than the forward solve and more suitable than general-purpose automatic differentiation (AD) for performance-critical applications. The utility of the simulator and its gradients is demonstrated in derivative-based optimization tasks, such as initial state estimation, and for training neural network controllers for optimal control.

**Strengths:**

- Non-trivial technical contribution: the paper presents a complete, end-to-end differentiable simulator for strong two-way rigid-fluid coupling using a Eulerian method. The analytical derivation of the adjoint pass for the entire pipeline, especially for the monolithic variational pressure solve.
- The resulting adjoint-based gradient computation is demonstrated to be exceptionally efficient. The authors report that the backward pass is faster than the forward solve and significantly outperforms general-purpose automatic differentiation frameworks like PhiFlow in runtime tests. This is promising, and it pushes in a direction that will benefit both the fluid dynamics and ICLR communities.
- The practical utility of the simulator is demonstrated on diverse optimization tasks. The computed gradients are successfully used for solving inverse problems and for training neural network policies for optimal control.

**Weaknesses:**

1) L064: It is not clear whether $\mathcal{L}$ is the objective terminal state, or the loss function that compares it to the predicted terminal stated. It appears to be described as the former, which is inconsistent with the typical machine learning literature.
2) Related work outlining a review of differentiable programming are fairly limited. One could think of [1, 2] among other studies
3) The method is limited to inviscid fluid simulation, limiting its applicability.
4) L304: the website link appears to be broken, or doesn't work. While the still frames are nice, they don't give an overview of the initial velocity which is often optimised. Having videos would significantly complement the results.
5) Figure 5 appears to show a single method, not two (including DiffFR) as claimed in L374.
6) The overview of the pipeline is interesting, but still limited. The Pseudo-code in Algorithm 1 would be much more useful if placed in the main text.

### Minor issues:
- L173: Figure ?

### References
- [ 1 ] Toshev et al., JAX-SPH: A Differentiable Smoothed Particle Hydrodynamics Framework, ICLR 2024 Workshop on AI4Differential Equations in Science
- [ 2 ] Nzoyem et al., A Comparison of Mesh-Free Differentiable Programming and Data-Driven Strategies for Optimal Control under PDE Constraints, SuperComputing Workshop on AI4Science

**Questions:**

1) L447: What steps were taken to ensure the Runtime Analysis against PhiFlow is fair ? Please provide details of this experiment, which is incredibly vague in the current manuscript ! (This is all the more concerning given that your code is in C++, while PhiFlow is in Python, which should definately be slower than yours)
2) What specifically makes the adjoint pass faster than the forward pass, as evidenced in Table 2. This is counter-intuitive, and the paper does not emphasize this enough. Please provide computationaal complexity analysis for Algorithms 2 and 3.
3) What are the main challenges in extending the proposed adjoint derivation to include viscous effects?
4) The paper claims the approach is highly scalable, but the experiments use relatively coarse grids (e.g., in L314). What is the computational and memory cost of this appraoch, and how far can you scale on the hardware avaialable?
5) L352: Specifically in the text optimisation, why was rotation not considered ?

---

> ### Author Response · Authors · 2025-11-20
> **Response to Weaknesses**
>
> We thank the reviewer for their valuable feedback and their questions.
>
> ## Weaknesses
>
> 1. **L064: It is not clear whether $\mathcal{L}$ is the objective terminal state, or the loss function that compares it to the predicted terminal stated. It appears to be described as the former, which is inconsistent with the typical machine learning literature.**
>
>     $\mathcal{L}$ is the notation for objective function, and $\mathcal{L} = ||q_o - q_t||^2$ where $q_o$ and $q_t$ are the observed and target states. There is a typo in the caption of Figure 1, and we apologize for this confusion.
> ---
> 2. **Related work outlining a review of differentiable programming are fairly limited. One could think of [1, 2] among other studies**
>
>     First of all, we thank the reviewer for bringing these works to our attention, and we apologize if we might have missed any others. We had to keep the available space in mind (especially with lengthy ICLR citation style) and limit it by favouring the depth of works that motivated our project, over the breadth of differentiable programming literature. Nevertheless, we will incorporate the suggested references and more to the final draft.
> ---
> 3. **The method is limited to inviscid fluid simulation, limiting its applicability.**
>
>     We feel that differentiable solid-fluid coupling with inviscid fluids is challenging enough, as evident from the complexity of the derivation of gradients and the general scope of our work in terms of complexity of design, implementation and experiments. And we disagree that our work has limited applicability, since there are a lot of fluids that can modelled without viscosity, and it forms the focus of a large volume of existing research work. And **we strongly feel that there are enough insights to gain from non-viscous solid-fluid coupling alone (both in terms of simulation in general and its differentiability) that it deserves a separate study**. Besides, **we show in the questions section below, how our work can be extended to support viscosity**.
> ---
> 4. **L304: the website link appears to be broken, or doesn't work. While the still frames are nice, they don't give an overview of the initial velocity which is often optimised. Having videos would significantly complement the results.**
>
>     The video results can also be accessed from the supplementary. We apologize for the broken link. We have fixed [the link](https://iclrauthor7856.github.io/DRiFT-Differentiable-Grid-Based-Rigid-Fluid-Coupling-for-Training-and-Control/) in the revised draft.
> ---
> 5. **Figure 5 appears to show a single method, not two (including DiffFR) as claimed in L374.**
>
>     We thank the reviewer for bringing this to our attention. We will include this to our revised draft.
> ---
> 6. **The overview of the pipeline is interesting, but still limited. The Pseudo-code in Algorithm 1 would be much more useful if placed in the main text**
>
>     The pseudocode was intended as a supplementary to the main explainer diagrams (Figure 1 and Figure 2), which is why we placed it in the appendix. We will shift Algorithm 1 to the main text if it better helps the exposition. **We would like to additionally request the reviewer to elaborate on why they feel the diagram of pipeline is limited and how it can be improved.**
> ---
> 7. **L173: Figure?**
>
>     Thank you for spotting the typo! We have fixed it in the revised draft.
>
> Because of lack of space, we address the questions in a separate comment below.

---

> ### Author Response · Authors · 2025-11-20
> **Response to Questions, Part 1**
>
> ## Questions
>
> 1. **What steps were taken to ensure the runtime analysis against PhiFlow is fair? Please provide details of this experiment, which is incredibly vague in the current manuscript! (This is all the more concerning given that your code is in C++, while PhiFlow is in Python, which should definitely be slower than yours)**
>
>     PhiFlow has data structures and operators to handle grid-based operations related only to fluids and static rigid bodies as obstacles, not two-way coupling. **We had to reimplement our two-way coupling method in Python on top of PhiFlow function calls** to ensure fairness in comparison. Additionally,
>
>     * We made sure to keep all parameters like grid resolution, cell width, time step, initial state, etc. same between experiments.
>
>     * The numerical algorithms employed and their settings (order of method, tolerance, iterations, etc.) like semi-lagrangian advection, conjugate gradient for pressure solve, are same in both cases.
>
>     * Parallelization wise, we had to rely on PhiFlow's internal GPU parallelization since we didn't find any option to choose. The PhiFlow examples were executed on an A5000 GPU node. Any code that we added for two-way coupling, was parallelized on CPU using concurrent.futures library and run on 16 threads, similar to our CPU parallelization. This means PhiFlow on GPU is slower than our code on CPU.
>
>     * In PhiFlow's auto-differentiation, the gradients were calculated only w.r.t. the target variable, i.e. the rigid body velocity, to avoid any overhead of calculating derivatives w.r.t. other parameters.
>
>     We will include these details and more in the revised draft. Even if we admit that there are inherent performance differences between Python and C++, note that **PhiFlow uses JAX, PyTorch or Numpy under the hood, all of which are highly optimized and use significant amount of C or C++ code as backend**. So, it is not entirely unfair to compare it with our C++ based simulator.
>
>     For more fair comparison, we propose the following empirical analysis. We can try to obtain the language overhead factor of Python over C++ by obtaining ratio $\alpha$ of forward pass times of PhiFlow and DRiFT. This is fair, since both forward passes implement the same simulation algorithm i.e. two-way solid-fluid coupling. Then we multiply DRiFT's adjoint pass time with $\alpha$ and compare the product with PhiFlow's autodiff time. **Note that by doing this, we are being less fair to our own method because PhiFlow, using PyTorch under the hood, is expected to be highly optimized while DRiFT is not, since that was not our main focus in this work.** So, any speedup obtained by this analysis is a lower bound on the actual speedup. Taking values from Table 2 of paper, we report the modified speedup below:
>
>     | **Experiment** | $\mathbf{\alpha}$ | ($\mathbf{\alpha \times}$ **DRiFT Adjoint Pass Time**) | **PhiFlow Autodiff Time** | **Normalized Speedup (Lower Bound)** |
>     | ---- | ---- | ---- | ---- | ---- |
>     | Dam Break 3D | 3.08 | 57.7 | 101.3 | 1.76 |
>     | Dam Break 2D | 5.33 | 0.59 | 0.98 | 1.66 |
>     | Text Optimization | 22.15 | 7.08 | 7.54 | 1.06 |
>     | Block Drop 2D | 5.83 | 0.23 | 0.39 | 1.67 |
>     | Block Drop 3D | 4.77 | 17.56 | 36.9 | 2.1 |
>
>     Evident from the above table, our method is faster even after (empirically) accounting for language differences. And note once again, **that the speedups reported in the last column are lower bounds on the achievable speedup**, should we manage to optimize our code even further. And finally, **we feel that what matters at the end is the final wall clock time and the speedup therein**.
> ---
> 2. **What specifically makes the adjoint pass faster than the forward pass, as evidenced in Table 2. This is counter-intuitive, and the paper does not emphasize this enough. Please provide computational complexity analysis for Algorithms 2 and 3.**
>
>     Please **refer to our response to the questions of reviewer gZxW above**, where we detail the computational complexity of our algorithm. In practice, majority of the time of the forward pass is taken by the marker particle advection (about 75\%) (for e.g. in the block drop exp., one frame in forward pass takes about 391ms out of which 297ms are spent in particle advection). This is reflected in the $O(PMT)$ term in the above complexity analysis. These marker particles are crucial for tracking and rendering the liquid surface. This is not required in the adjoint pass, as we store the liquid SDF computed in the forward pass and reuse it. This is where the difference in runtime occurs.

---

> ### Author Response · Authors · 2025-11-22
> **Response to Questions, Part 2**
>
> 3. **What are the main challenges in extending the proposed adjoint derivation to include viscous effects?**
>
>     In Navier Stokes equation, viscosity appears as an additional term $\frac{1}{\rho} \nabla \cdot [\nu (\nabla u + (\nabla u)^T)]$ [1], and if we stick with operator splitting, **all we need to do to extend our simulator is introduce a viscosity stage after the pressure solve**. In this stage, we can use a variational interpretation of viscosity [1] to convert the solid-fluid coupling PDE into a **linear system in combined solid-fluid velocity** [2]. But using this approach, the velocities obtained at the end of the time step are not divergence free, requiring an additional pressure projection step. This is undesirable since projection is numerically dissipative. Newer methods like [3] solve both the pressure and viscosity coupled in one giant monolithic solve. **In both cases, the underlying dynamics result in a linear system, so adjoint method can be adapted naturally**. **The main challenge is deriving the gradient of the solution operator with respect to rigid body positions**, (i.e. the term $\mathbf{\frac{\partial J}{\partial x}}$ in our paper). **In summary, solving for viscosity is a straighforward extension by design of our pipeline, but due to technical details, implementation overhead and numerical issues that pop up, we believe it is better to address it in a separate, future work**.
>
>     ### References
>
>     [1] Batty et al. Accurate Viscous Free Surfaces for Buckling, Coiling, and Rotating Liquids
>
>     [2] Takahashi et al. A Geometrically Consistent Viscous Fluid Solver with Two-Way Fluid-Solid Coupling
>
>     [3] Takahashi et al. Monolith: A Monolithic Pressure-Viscosity-Contact Solver for Strong Two-Way Rigid-Rigid Rigid-Fluid Coupling
> ---
> 4. **The paper claims the approach is highly scalable, but the experiments use relatively coarse grids (e.g., in L314). What is the computational and memory cost of this appraoch, and how far can you scale on the hardware available?**
>
>     Once again, **we refer the reviewer to our response to the questions of reviewer gZxW above** for the detailed complexity analysis of our algorithm. Before proceeding to our calculation of scalability, we would like to preface it by saying that although we do an asymptotic analysis earlier, the constant factors are significant but tedious to calculate, so the actual results of the following analysis might vary considerably, but should stay in the same order of magnitude. That being said, **for just the forward simulation, the peak memory usage per frame** is $O(MN + P)$, which in exact terms is $6MN + 30N$, since we have 6 DOFs in 3D we use roughly 10 particles per grid cell with 3 DOFs each. Assuming we have only one rigid body and we use single precision (32-bit) variables, total memory consumption is $1152N$ bits. Assuming 32GB RAM, this **allows us to simulate about $\mathbf{2.2 \times 10^8}$ grid cells**. On the other hand, **for the entire optimization pipeline (forward pass + adjoint pass)**, we also store the simulation state at the end of every frame of forward pass, to avoid recomputation in the adjoint pass. The **peak memory usage**, which occurs at the end of the forward pass, scales as $O(F \times (M + N))$, which in exact terms is about $6FM + 6FN$, again 6 DOFs for each rigid body, and 6 grids for fluid (3 for fluid velocity, one for pressure, one each for liquid and solid SDF). Again, assuming only one rigid body, this scales approximately as $192FN$ bits. Again, assuming 32GB RAM, this allows us to simulate $\mathbf{\frac{1.33 \times 10^{8}}{F}}$ grid cells, where $F$ is the number of simulation frames. For e.g. 100 frames allows us to simulate a maximum of $1.33 \times 10^6$ grid cells and so on.
>
>     In conclusion, **our simulator supports higher resolution simulations**, it's just that we felt the chosen resolutions were sufficient for justifying our contribution and proving our point. But **if the reviewer feels any additional specific high-resolution scenario is necessary, we are glad to perform it and add it to the final version**.

---

> > ### Comment · Reviewer_Nvep · 2025-11-26
> >
> > Dear Authors, thank you for addressing my concerns. I am grateful for your thorough rebuttal.
> >
> > The weaknesses I pointed have been fully addressed. That said, I still have a few comments regarding the questions:
> >
> > 1) This is a very interesting experiment. Although there is a case to be made that PyTorch is at a disadvantage given the grid-based (and not Tensor-base) simulations. Frameworks like Nvidia's WARP [1] are built to address this exact issue.
> > 2) Thanks for this explanation. It is clear now. I would suggest including it in a prominent section of the main text.
> > 3) Thanks for this as well. It is understandable that this would be a separate paper.
> > 4) Thank you for your analysis, but one single large-scale experiment would have been satisfactory (assuming compute is available, and that you have enough rebuttal time left). To be clear, a simple *scaling analysis* (showing speedup vs resolution) would significantly bolster this work.
> >
> > I understand if there's not time left to do the additional experiment in 4).
> >
> > [1] https://github.com/NVIDIA/warp

---

> ### Author Response · Authors · 2025-12-01
> **Response to Comments**
>
> We thank the reviewer for going through our rebuttal and we are glad that they find our response satisfactory. Here, we respond to their follow-up questions and comments:
>
> 1. **Although there is a case to be made that PyTorch is at a disadvantage given the grid-based (and not Tensor-base) simulations. Frameworks like Nvidia's WARP [1] are built to address this exact issue.**
>
>     We would like to clarify that PhiFlow internally stores all grids as PyTorch tensors. So, all fluid based operations (velocity advection, particle advection, enforcing incompressibility) use the optimized tensor operations. Only the two-way coupling part that we implement on top of PhiFlow for comparison is grid-based where we use grids for computation. So, PyTorch is not entirely at a disadvantage. And whatever overhead (disadvantage) we introduce in our own implementation through these grid-based operations, we factor it out as $\alpha$ in our empirical experiment above. We also considered using Warp for comparison, but we finally decided against it, because of the implementation overhead.
> ---
> 2. **I would suggest including it in a prominent section of the main text.**
>
>     We have included this discussion in **Runtime Analysis in Section 6.3 and Appendix G** in the updated paper.
> ---
> 3. **A simple scaling analysis (showing speedup vs resolution) would significantly bolster this work.**
>
>     We thank the reviewer for suggesting this experiment. We took the 2D Dam Break experiment and scaled the resolution by factors of 2 in each dimension upto 4 (since we run out of memory in scaling any further). We keep the domain size constant by adjusting the grid cell width $\Delta x$. We appropriately adjust the time step $\Delta t$ to obey the CFL condition, and the number of frames $F$ to keep the simulation length at 1s. After running these scenarios in our simulator and PhiFlow, we note the average runtimes of one forward and backward pass and evaluate the speedup. These **results are summarized in the table below and Appendix G**. We find that the speedup decreases as the resolution increases. This might be due to the fact that our method is CPU parallelized predominantly and uses a maximum of 16 threads. While PhiFlow uses GPU parallelization, and thus it can leverage more cores at higher resolutions for effective concurrency. We identify this as one of our limitations, and we have updated the limitations section to account for this.
>
>     | Resolution      | $\Delta x$        | $\Delta t$           | $F$  | Ours (Forward $\Phi$) | PhiFlow (Forward $\Phi$) | Ours (Adjoint $\Psi$) | PhiFlow (Adjoint $\Psi$) | Speedup in Adjoint Pass |
>     |-----------------|-------------------|-----------------------|------|------------------------|----------------------------|-------------------------|---------------------------|---------|
>     | $35 \times 20$  | $10^{-1}$         | $2 \times 10^{-2}$    | $50$ | $5.05 \times 10^{-2}$  | $4.8 \times 10^{-1}$       | $2.13 \times 10^{-2}$   | $5.16 \times 10^{-1}$     | $24.22$ |
>     | $70 \times 40$  | $5 \times 10^{-2}$| $10^{-2}$             | $100$| $1.74 \times 10^{-1}$  | $7.86 \times 10^{-1}$      | $1.21 \times 10^{-1}$   | $9.57 \times 10^{-1}$     | $7.91$  |
>     | $140 \times 80$ | $2.5 \times 10^{-2}$| $5 \times 10^{-3}$  | $200$| $1.36$                 | $2.05$                     | $1.49$                  | $2.71$                    | $1.82$  |
>     | $280 \times 160$| $1.25 \times 10^{-2}$| $2.5 \times 10^{-3}$| $400$| $4.30$                 | $6.59$                     | $18.46$                 | $9.72$                    | $0.53$  |

---

### Official Review · Reviewer_gZxW · 2025-10-30

**Soundness:** 3
**Presentation:** 2
**Contribution:** 2
**Rating:** 4
**Confidence:** 4

**Summary:**

This paper presents a grid-based two-way coupled differentiable fluids solver. The authors implement an adjoint method for computing control variable updates relative to a user-defined reward function. One of the contributions that the authors present is a derivation of analytic gradients for computing updates for the gradients of the pressure solve, that affect both the fluid and the coupled rigid body. The authors also use a variational formulation for the fluid-solid boundary, and a ghost fluid method for the fluid-air interface. Their results demonstrate that the proposed approach works reasonably for a set of predefined low-resolution guidance tasks.

**Strengths:**

- Working with fluids optimization is a though problem. It usually slow, hard to debug, requires a lot of memory, boundary conditions can be tricky, and the method has to be implemented precisely in all its stages to produce correct results.
- The proposed variational approach and the ghost fluid discretization for liquid-air interfaces are solid modelling choices.
- The derivation of the analytical updates seem correct, and the results demonstrate that the method works.

**Weaknesses:**

- The quality of the results is sub-par. Grid resolutions are really coarse and there are not a lot of different examples. Also hard to really evaluate the method, since I was not able to access the website with the video results.
- The method does not seem efficient. DiffFR uses 237699 particles and it is 2.3 times slower than the proposed method, but the authors compare against a fairly coarse grid setup of 39x15x24 = 14k cells. Therefore the proposed method uses about 5% of the variables of DiffFR and its x2 faster than it.
- A first-order (Semi Lagrangian advection with forward Euler) fluid solver is an outdated approach. Hybrid (grid + particles), impulse-based methods are more effective and precise.
- Missing references: "Honey, I Shrunk the Domain: Frequency-aware Force Field Reduction for Efficient Fluids Optimization", "Efficient Solver for Spacetime Control of Smoke".
- Minor typos
   - On plots, whats the concept of Epochs? Shouldn't it be iterations?
   - L173: Figure ??
   - L107: Langrangian

**Questions:**

I would like to understand what are the computational limitations of the proposed approach. How the memory and time would scale with grid size and what would be effective ways to deal with longer simulations.

---

> ### Author Response · Authors · 2025-11-19
> **Response to Weaknesses**
>
> We thank the reviewer for their valuable feedback.
>
> ## Weaknesses
>
> 1. **The quality of the results is sub-par. Grid resolutions are really coarse and there are not a lot of different examples. Also hard to really evaluate the method, since I was not able to access the website with the video results.**
>
>     The video results can also be accessed from the supplementary. We apologize for the broken link. We have fixed [the link](https://iclrauthor7856.github.io/DRiFT-Differentiable-Grid-Based-Rigid-Fluid-Coupling-for-Training-and-Control/) in the revised draft. The **demo examples were chosen to showcase different aspects of solid-fluid coupling**:  Rigid body drop (Figure 5) to show effect of static liquid on rigid body, text optimization (Figure 4) to extend to multiple rigid bodies, dam break (Figure 3) for full two-way coupling, and optimal control (Figure 7) to show utility of our simulator in training neural networks. We believed this to be sufficient to justify our contribution. Similarly, **our simulator supports higher resolution simulations**, it's just that we felt the chosen resolutions were sufficient for proving our point. But **if the reviewer feels any additional specific high-resolution scenario is necessary, we are glad to perform it and add it to the final version**. If by sub-par, the reviewer refers to visual quality, we admit that our main focus was not on graphics but to adequately demonstrate the optimization procedure.
> ---
> 2. **The method does not seem efficient. DiffFR uses 237699 particles and it is 2.3 times slower than the proposed method, but the authors compare against a fairly coarse grid setup of 39x15x24 = 14k cells. Therefore the proposed method uses about 5\% of the variables of DiffFR and its x2 faster than it.**
>
>     We apologize for this but the runtimes reported are older numbers. Current runtimes: **forward simulation: 6.58s**, **adjoint simulation: 3.68s**, **total optimization time over 50 epochs: 8.6 minutes**, **speedup:** $\mathbf{19.9\times}$ (check **Table 2, Runtime Analysis**). We will update this in the DiffFR section. We understand this may not be enough considering the large difference between degrees of freedom, we list possible reasons:
>
> * DiffFR doesn't compute the full gradient to avoid stability issues, while we compute the full gradient without stability issues. This is evident in our much lower final loss values after 50 epochs (Figure 6c).
>
> * About 75\% of our forward pass (391ms per frame) is spent in advecting marker particles (297ms per frame) which is not a strictly grid-based operation. Optimizing this can yield significant gains, but we leave this to future work, since in the current version, marker particles are important for tracking the liquid surface.
>
> * We don't compete with Lagrangian methods like DiffFR for accuracy, so we believe it **isn't exactly fair to compare times at similar degrees of freedom**. The whole **point of using grid-based methods is that we can forego some accuracy (i.e. degrees of freedom) to save computation time, even if the gain is not proportional to accuracy sacrificed**, provided the accuracy is within threshold.
> ---
> 3. **A first-order (Semi Lagrangian advection with forward Euler) fluid solver is an outdated approach. Hybrid (grid + particles), impulse-based methods are more effective and precise.**
>
>     Thank you for bringing this point in discussion as we believe this is an important future thread of research. The **advection step**, though not being very complex, **has the highest contribution in error in gradient computation** (Table 1). We believe this is because of bad conditioning of the matrix $\mathbf{W} + \frac{\partial \mathbf{W}}{\partial {u}} {u}_t$ in eqn (3). More the number of velocity samples on which velocity of current time step depends, worse the conditioning of the matrix. So, **a second order method like RK2 would introduce more instability in gradient computation**. This is the reason we use a first-order method for now.
>
>     We think hybrid methods like MPM are not suitable for rigid-fluid coupling, as **MPM suffers from enforcing the rigidity constraint on the solid, or strict incompressibility constraint on the liquid**. **Impulse-based methods are explicit and partitioned**, and thus support only **weak-coupling**, which is **prone to stability issues and violation of boundary conditions**, especially at solid-fluid boundary.
> ---
> 4. **Missing references**
>
>     Thank you for bringing these to our attention! We have included them in the revised draft.
> ---
> 5. **On plots, whats the concept of Epochs? Shouldn't it be iterations? + Typos**
>
>     We consider one epoch as one forward pass + loss computation + backward pass + gradient descent. We didn't use 'iteration' to avoid confusion with iterations (frames) in a simulation. And thank you for spotting the typos! We have fixed them in the revised draft.
>
>
> Because of lack of space, we address the question in a separate comment below.

---

> ### Author Response · Authors · 2025-11-19
> **Response to Questions**
>
> ## Questions
>
> 1. **I would like to understand what are the computational limitations of the proposed approach. How the memory and time would scale with grid size and what would be effective ways to deal with longer simulations.**
>
>     Suppose the number of grid cells is N, number of marker particles is P and there are M rigid bodies with maximum T triangles each. And let the number of frames in an epoch be F.
>
>     ### **Memory**
>
>     * Fluid State, $O(N + P)$: Velocity, $O(N)$ + Pressure, $O(N)$ + SDF (signed distance function), $O(N)$ + Marker Particles, O(P)
>     * Rigid Body State, $O(M + N)$: Velocity, $O(M)$ + Position, $O(M)$ + SDF, $O(N)$
>     * Combined State, $O(M + N + P)$
>     * Sparse Gradient Matrix **G**, $O(N)$
>     * Solid-Fluid Coupling Matrix **J**, $O(NM)$
>     * One iteration(frame) of forward pass, $O(MN + P)$
>     * One iteration(frame) of backward pass, $O(MN)$: No marker particles in backward pass
>     * One optimization epoch (forward + backward pass), $O(F \times (M+N))$: Because marker particles and temporary matrices are discarded after every frame
>
>     ### **Time (unparallelized)**
>
>     * Advection, $O(N)$
>     * Particle Advection + Fluid SDF Computation, $O(PMT)$
>     * Rigid Body Update, $O(M)$
>     * Rigid Body SDF Computation, $O(MT)$
>     * Pressure Solve, $O(NM + (N + M)^{1.5})$: Matrix constructions, $O(NM)$ + Linear solve using conjugate gradients, $O((N+M)^{1.5})$
>     * Velocity Correction, $O(N + M)$: sparse matrix-vector multiplication of size $M + N$
>     * One iteration(frame) of forward pass, $O(NM + (N + M)^{1.5} + PMT)$
>     * One iteration(frame) of backward pass, $O(NM + (N + M)^{1.5})$
>     * One optimization epoch (forward + backward pass), $O(F \times (NM + (N + M)^{1.5} + PMT))$
>
> * We can save some memory per frame by using sparser **J** to bring down its space to $O(N)$, but this is not significant in a longer simulation since it is discarded anyway at the end of the frame. A more effective startegy would be to not store any intermediate frames, but use **checkpointing** for backward pass. This brings down the memory requirements to O(M + N) but increases the computation time of the backward pass by an equivalent factor.
>
> * For a fixed background grid, the above reported times are the best possible times (on a single thread/core) as far as we are aware. In practice, the particle advection (i.e. $O(PMT)$) takes the most time. So, one way is to optimize marker particle advection or employ a grid-based advection scheme like level-set advection. Another way this can be made much faster is if we use **adaptive grids**, which we are currently working on. Adpative grids have finer resolution in areas requiring higher accuracy (i.e. near solid-fluid boundary) and coarser resolution in other macroscopic areas (i.e. fluid regions). Such grids have the potential to be efficient both in space and time. But differentiability is a challenge in adaptive grids because of dynamic size of state variables.
>
> We leave both these optimizations to future work.

---

### Official Review · Reviewer_U6dN · 2025-10-31

**Soundness:** 2
**Presentation:** 3
**Contribution:** 2
**Rating:** 6
**Confidence:** 2

**Summary:**

This paper presents DRiFT, a differentiable grid-based fluid simulation with two-way coupling between fluid and rigid bodies. The grid-based discretization enables analytically deriving the gradients over each phase of the fluid simulation, covering fluid velocity advection, pressure solve for boundary conditions, rigid body update, and velocity correction for boundary conditions. The authors demonstrate that gradients from DRiFT can be used for simple optimization tasks, including training a neural network to predict a control sequence. DRiFT obtains 5-10x speedup over a comparable Eulerian differentiable fluid simulation that uses auto-differentiation from PyTorch to compute gradients.

**Strengths:**

- The authors demonstrate how to analytically derive gradients for grid-based fluid simulation. They demonstrate that the gradients computed by adjoint-based gradient computation can be used for optimization of the initial state to achieve some desired final state. given some cost function, and to optimize control forces by training some neural-network based controller to predict a control sequence.

- The proposed differentiable fluid simulation considers strong two-way coupling between fluid and rigid bodies. I believe most prior work on differentiable soft-body simulation is only capable of one-way coupling.

- The authors release their differentiable fluid simulation code, which benefits reproducibility.

**Weaknesses:**

- Based on the videos in the supplementary, the simulated fluid appears highly viscous. How stable is the simulation over longer simulation times? How accurate is the fluid simulation compared to Aquarium, PhiFlow, or other methods for fluid simulation, given that the proposed method uses a grid-based discretization?

**Questions:**

It would be helpful to discuss hybrid Eulerian-Lagrangian approaches for fluid simulation such as the Material Point Method (MPM) used in differentiable simulators Fluidlab [1], DaXBench [2], or Rewarped [3]. For instance, I believe MPM easily handles simulating different types of fluids in the same scene. How easy is it to extend DRiFT to multiple fluids?

[1] https://arxiv.org/abs/2303.02346

[2] https://arxiv.org/abs/2210.13066

[3] https://arxiv.org/abs/2412.12089

Line297: Does DRiFT support parallel simulation? Or is OpenMP/CUDA only used to parallelize operations in a single physics scene?

Figure 4: Choice of epochs to visualize seems arbitrary (39, 58, 19). How stable is the optimization procedure? Running simulations with different random seeds or number of iterations and including error bars for Figure 6 would be helpful.

Section 6.2: Are the control forces executed in an open loop, i.e the policy predicts the entire control sequence?

- - -

[minor]

Line173: Missing figure number reference

Figure 4: Should be epoch 19 not 10?

---

> ### Author Response · Authors · 2025-11-18
>
> We thank the reviewer for their valuable feedback and their questions.
>
> ## Weaknesses
>
> 1. **Why does the simulated fluid appear highly viscous in the videos? How stable is the simulation over longer simulation times? How accurate is the simulation compared to Aquarium, PhiFlow, or other methods, given that the proposed method uses a grid-based discretization?**:
>
>     We use an implicit method for **solving for fluid pressure using kinetic energy minimization**, which is inherently **dissipative**. So, over simulations long enough, the combined solid-fluid motion would settle down to an energy minimum, which corresponds to an equilibrium visually. But, this makes our method **very stable**. We have tried simulating for hundreds (and even upto a few thousand sometimes) of frames without suffering from numerical explosion issues. But, our gradient computation (because of advection, details in response to Reviewer gZxW, Weakness 3) diverges from the finite difference gradient as number of frames increase. Note that the **gradient computation method is itself unconditionally stable, just not accurate for longer simulations**. This is especially problematic since we expect the loss landscape of solid-fluid coupling to be highly non-convex. We do not explicitly model any viscous effects, so any **visual viscosity can either be due the numerical dissipation explained above, or might be because of post-processing issues like surface reconstruction (for which we use a third party library called SplashSurf), framerate of the video, etc.** We are not entirely sure regarding this matter. We are in the process of obtaining results for accuracy comparisons with PhiFlow and Aquarium, and we will post them here and in the paper as soon as we obtain them.
> ---
> ## Questions
>
> 1. **It would be helpful to discuss hybrid Eulerian-Lagrangian approaches such as Material Point Method (MPM) used in differentiable simulators Fluidlab, DaXBench, or Rewarped. For instance, I believe MPM easily handles simulating different types of fluids in the same scene. How easy is it to extend DRiFT to multiple fluids?**
>
>     We are not entirely sure how much effort would be required to support multiple fluids. At the minimum, **we would need to model and implement the interaction dynamics (and their derivatives)** between different fluids, not to mention accompanying **challenges like tracking multiple domains, volume fractions, boundary conditions**, etc. We would say this is currently out of scope of our contribution. We agree that hybrid Lagrangian-Eulerian methods like MPM can handle multiple fluids. But our main goal is rigid-fluid coupling and **MPM suffers from enforcing the rigidity constraint on the solid, or strict incompressibility constraint on the liquid**. Besides, our method is also hybrid Eulerian-Lagrangian in a different sense, using Eulerian dynamics for liquid and Lagrangian dynamics for rigid body. However, the coupling dynamics are grid-based. In any case, thank you for bringing these works to our attention. We will include them and the above discussion in the revised draft.
> ---
> 2. **Line297: Does DRiFT support parallel simulation? Or is OpenMP/CUDA only used to parallelize operations in a single physics scene?**
>
>     The current version of our simulator only supports operation level parallelization, accelerating only a single simulation with a fixed initial state. But, it is easy to extend our work to support multiple parallel simulations by appropriately vectorizing the state variables and assembling solution operators block-wise.
> ---
> 3. **Figure 4: Choice of epochs to visualize seems arbitrary (39, 58, 19). How stable is the optimization procedure? Running simulations with different random seeds or number of iterations and including error bars for Figure 6 would be helpful.**
>
>     We choose the epoch with the smallest value of loss $\mathcal{L}$ to visualize. Our method is deterministic for the most part (except the marker particle initialization at t = 0, which in our experience does not influence the simulation in later time frames). Meaning that for a given initial state, our forward and adjoint simulations will be exactly same on different runs. The optimizer we use is also deterministic (Gradient Descent or Adam). So, overall our results do not change on different runs. This is why we don't report any error bars in any experiment in Section 6.1.
> ---
> 4. **Section 6.2: Are the control forces executed in an open loop, i.e the policy predicts the entire control sequence?**
>
>     According to your definition, the **control force is open loop**, meaning given the initial state, the controller generates the control force for the entire sequence at once. But the **optimization is closed loop**, meaning the differentiable simulator generates gradients for the entire trajectory as feedback for training the controller.
> ---
> 5. **Typos**
>
>     Thank you for spotting these! We have fixed them in the revised draft.

---

> > ### Comment · Reviewer_U6dN · 2025-11-26
> >
> > > visual viscosity can either be due the numerical dissipation explained above, or might be because of post-processing issues like surface reconstruction ...
> >
> > Right, seems like this is an artifact of having to perform visual post-processing for grid-based approaches to fluid simulation similar to MPM. The fluid could appear less viscous with smaller cell sizes given the same grid dimensions or larger grid dimensions?
> >
> > > We choose the epoch with the smallest value of loss $\mathcal{L}$ to visualize.
> >
> > What if you run the optimization for 10x more iterations? Given the performance optimizations described in L15, each experiment takes under 1 hour to run currently? How stable is the result?
> >
> > > Our method is deterministic for the most part ... Meaning that for a given initial state, our forward and adjoint simulations will be exactly same on different runs.
> >
> > What if you apply some minor perturbations to the initial state? Given the sensitivity of physics simulations (and the claim that the proposed method is unconditionally stable), I think it would be useful to provide more empirical evidence to demonstrate the stability of the method across different runs.

---

> ### Author Response · Authors · 2025-12-02
> **Response to Comments**
>
> We thank the reviewer for going through our rebuttal. We address the follow-up questions below:
>
> 1. **The fluid could appear less viscous with smaller cell sizes given the same grid dimensions or larger grid dimensions?**
>
>     We believe there are two things to address for superior visual quality of liquid surface: amount of visual detail and the rate at which it this detail changes across frames (i.e. visual viscosity). Using a higher grid resolution (which is what we assume you mean when you say 'larger grid dimension') only addresses the visual detail. Using a higher frame rate to render the video would better help the visual viscosity. The smoothing kernel which is used to reconstruct liquid surface from marker particles affects both, since smoothing averages over the surface details in both space and time, by adding volume artificially.
> ---
> 2. **What if you run the optimization for 10x more iterations? Given the performance optimizations described in L15, each experiment takes under 1 hour to run currently? How stable is the result?**
>
>     We thank the reviewer for suggesting this experiment. First of all, we would like to clarify the approximate runtimes of *one experiment* of our demos of Section 6.1 (check **Table 2, Runtime Analysis** in the paper):
>
>     | **Demo** | **Frames per Epoch** | **Epochs** | **Total Time** |
>     |----------|----------|----------|----------|
>     |  Dam Break 3D  | 100  | 50 |  63.75 hours |
>     |  Dam Break 2D | 100  | 50 |  21.7 minutes |
>     |  Text Optimization | 100  | 20  | 21.7 minutes |
>     |   Block Drop 2D | 50 | 50 | 4.2 minutes |
>     |  Block Drop 3D  | 50 | 50 | 7.1 hours |
>
>     That being said, we rerun the five demos of section 6.1, but with $10 \times$ more epochs. But **because of the large amount of time that the 3D dam break experiment takes, we could only simulate it upto 50 frames in the given rebuttal period. We will include an extended optimization trajectory in the camera-ready version of the paper**. The results have been added to **Appendix E and Figure 9** of the updated paper. We find that the optimization trajectory of most of the experiments is stable, except that of the 2D dam break. But, we believe that might be because of the complex loss landscape and our simple optimizer (fixed learning rate gradient descent), rather than the accuracy and stability of our gradient computation. In any case, the loss value, even though oscillating, remains on average between $10^{-3}$ and $10^{-2}$.
> ---
> 3. **What if you apply some minor perturbations to the initial state?**
>
>     We once again thank the reviewer for suggesting this experiment. Once again, we rerun all five demos of Section 6.3 for one epoch, but now with small perturbations $\delta$ to the initial position and velocity of the rigid body. We obtain the average output perturbations $\Delta q = (\Delta u, \Delta x, \Delta v)$ component wise and list them in **Table 3, Appendix F (we avoid including the table here because of the its large size)**. We also plot the instability factor $\frac{\Delta q}{\delta}$ to see how the output perturbation changes relative to input perturbation. These are included in **Figure 10, Appendix F**. **Our conclusion is that the output perturbation is low is general and decreases with a decrease in input perturbation. But the rate of this decrease is lower for more complex scenarios like letter optimization or dam break.** Meaning that the more complex the simulation is, the higher the inherent inaccuracy in the simulation, so higher the output perturbation regardless of the magnitude of input perturbation. This translates into higher instability factor values.

---

### Author Response · Authors · 2025-11-24
**Response for Common Concerns**

We thank the reviewers for their valuable feedback and insightful questions! Here we reiterate and answer some common concerns:

1. **Videos for demos**: The website link for videos in the original link was inaccessible, and we are extremely sorry for that. Please find the   [new link here](https://iclrauthor7856.github.io/DRiFT-Differentiable-Grid-Based-Rigid-Fluid-Coupling-for-Training-and-Control/). The videos and code can also be found in the supplementary. Additionally, we have fixed the link in the updated paper.

2. **Related Work**: We thank the reviewers for bringing these works to our attention. We have incorporated these works and expanded our related works section.

3. **Computational Complexity Analysis**: We have included details of the asymptotic time and memory taken by our simulation algorithm, accompanied by possible ways to improve them, **in our response to Reviewer gZxW**. This is added in Appendix D in the updated paper.

---

### Author Response · Authors · 2025-12-02
**Summary of Rebuttal**

We thank all the reviewers for reading and analyzing our work, providing valuable feedback and raising insightful questions. We have addressed each of them to the best of our ability. Here, we summarize our responses, additional experiments we performed, updates to the paper and reviewer responses.

## Additional Experiments

| Experiment | Location | Our Conclusion |
| --- | --- | --- |
| Optimization Stability | Figure 9 and Appendix E | Our simulator provides stable gradients for longer optimization trajectories and any short term instability such as oscillations can be attributed to the complexity of the loss landscape and/or our choice of optimizer. |
| Simulation Stability | Figure 10, Table 3 and Appendix F | Our simulator is mostly stable with respect to input perturbations, but the rate of decrease in magnitude of output perturbations decreases with decrease in magnitude of input perturbations. We believe this might be due to a minimum inherent inaccuracy in grid-based methods, combined with unmodelled effects like interaction of rigid bodies with boundaries and other rigid bodies. |
| Scalability Analysis | Table 4 and Appendix G | Our gradient computation method is faster than PhiFlow's autodifferentiation at lower resolutions, with the speedup decreasing at higher resolutions, which we believe might be because of our predominantly CPU based parallelization vs PhiFlow's GPU parallelization |
| Empirical Analysis of Language Overhead | Response to Reviewer Nvep (Questions, part 1) | Even after we factor for language differences between C++ and Python (while putting our method at a disadvantage), the normalized speedup of our method over PhiFlow is greater than 1 |

## Updates to Paper
On top of the experiments mentioned above, we have added/modified the following things in the paper.

1. **Videos:** The anonymous [link to the videos](https://iclrauthor7856.github.io/DRiFT-Differentiable-Grid-Based-Rigid-Fluid-Coupling-for-Training-and-Control/) of demos is fully functional now. The videos can also be accessed in the supplementary.
2. **Computational Complexity Analysis:** We have included a comprehensive complexity analysis of our method and ways to improve it in **Appendix D**. Based on this analysis, we also explain in Section 6.3, Runtime Analysis how our gradient computation (adjoint pass) is faster than the forward pass at lower resolutions.
3. **DiffFR results**: We have added image results for our particle based baseline DiffFR for the rigid drop demo.
4. **Comparison with MPM**: We have expanded our related works section, Section 2, to include a discussion of hybrid Lagrangian-Eulerian methods like MPM and how they are unsuitable at the moment in our opinion for strong two-way coupling.
5. **Pseudocode**: We have moved the pseudocode from Appendix to the main text to better help the exposition.

## Reviewer Responses

1. **Reviewer U6dN**: The reviewer seems to agree with our response that the visual viscosity in the video results might not be a weakness of the simulation method but post-processing like surface reconstruction and frame-rate. We feel we have answered all the questions they initially raised, and performed all experiments (Appendix E and F) they suggested in their follow-up comments.

2. **Reviewer gZxW**: The reviewer pointed to three main weaknesses:

    * **Video results and higher resolution scenarios**: In response, we have provided the link for video results and performed scalability analysis in Appendix G for higher resolution experiments.
    * **Efficiency of our method in comparison with DiffFR**: We clarified that our method is actually $20 \times$ faster which we misreported earlier (check Table 2 for runtimes) which is closer in proportion of degrees of freedom employed in both methods. Further, we give possible reasons why DiffFR might be faster in proportion and how it might not be fair to compare runtimes at similar degrees of freedom.
    * **Higher order advection**: We explain how the gradient computation might be unstable for higher order advection methods.

    Additionally, we provide the computational complexity analysis mentioned earlier as a response to this reviewer's question. However, they haven't had a chance to respond to our rebuttal.

3. **Reviewer Nvep**: The reviewer feels that we have addressed all the weaknesses they raised of our method. Specifically, we explained how we ensured fairness in comparing our method with PhiFlow. Then, we provided technical details of how our method can be extended to support viscosity, and how it is out of the scope of our current work. With the help of our complexity analysis, we explained how our adjoint pass is faster than our forward pass at lower resolutions, which the reviewer finds prominent. Finally, to address the reviewer's follow-up regarding scalability and higher resolution scenarios, we performed the aforementioned scalability analysis (Appendix G).

---

### Meta-Review · Area_Chair_shzx · 2026-01-06

**Summary:**

This paper proposes a differentiable, particle-based simulation framework with efficient gradient computation for rigid-fluid coupled system. The reviewers raised major concerns regarding the practical advantages, scalability, and comparative efficiency of the proposed approach relative to existing methods. Although the rebuttal adds new experiments and clarifications, these additions do not fully resolve the core issues that motivated skepticism in the initial reviews.

**Reviewer Concerns:**

The authors provide additional experiments on optimization stability, robustness to initial state perturbations, and scalability, and improve the clarity of the presentation through added complexity analysis, pseudocode, and demo videos. These efforts strengthen the paper and address several concerns.

However, key issues remain unresolved. The efficiency advantage of the proposed method over existing differentiable simulators is not convincingly demonstrated, particularly in comparison with DiffFR, where performance gains diminish at higher resolutions and remain sensitive to implementation choices and language differences. As a result, the practical significance of the claimed speedups is unclear.

Moreover, several important limitations are acknowledged but left unaddressed, including restrictions related to higher-order advection, viscosity modeling, and broader physical interactions. While these may be out of scope, they limit the applicability of the method to realistic simulation and optimization tasks. Overall, despite the additional experimental evidence, the improvements appear incremental relative to existing frameworks, and the paper does not yet provide a sufficiently compelling case for acceptance.

**Reviewer Scores:**

Reviewer U6dN is likely to keep the score 6 or raise to 8. The visual viscosity was partially clarified. The authors also provide numerical results with longer optimization and initial state perturbation to demonstrate the robusntess of the proposed optimization scheme.

Reviewer gZxW may keep the score 4 given that efficiency comparisons remain unconvincing.

Reviewer Nvep may increase the score from 2 to 4 or 6 since he indicated that the weaknesses were fully addressed.

---

### Decision · Program_Chairs · 2026-01-26

Reject